# ROBUST ROOT CAUSE DIAGNOSIS USING IN-DISTRIBUTION INTERVENTIONS

**Lokesh Nagalapatti**[1]*, **Ashutosh Srivastava**[2], **Sunita Sarawagi**[1], **Amit Sharma**[3]
[1]Indian Institute of Technology Bombay
[2]International Institute of Information Technology Hyderabad
[3]Microsoft Research India

## ABSTRACT

Diagnosing the root cause of an anomaly in a complex interconnected system is a pressing problem in today's cloud services and industrial operations. We propose In-Distribution Interventions (IDI), a novel algorithm that predicts root cause as nodes that meet two criteria: 1) *Anomaly:* root cause nodes should take on anomalous values; 2) *Fix*: had the root cause nodes assumed usual values, the target node would not have been anomalous. Prior methods of assessing the fix condition rely on counterfactuals inferred from a Structural Causal Model (SCM) trained on historical data. But since anomalies are rare and fall outside the training distribution, the fitted SCMs yield unreliable counterfactual estimates. IDI overcomes this by relying on interventional estimates obtained by solely probing the fitted SCM at in-distribution inputs. We present a theoretical analysis comparing and bounding the errors in assessing the fix condition using interventional and counterfactual estimates. We then conduct experiments by systematically varying the SCM's complexity to demonstrate the cases where IDI's interventional approach outperforms the counterfactual approach and vice versa. Experiments on both synthetic and PetShop RCD benchmark datasets demonstrate that IDI consistently identifies true root causes more accurately and robustly than nine existing state-of-the-art RCD baselines. Code will be released at https://github.com/nlokeshiisc/IDI_release.

## 1 INTRODUCTION

In recent years, cloud services have become increasingly popular due to their advantages in resource sharing, scalability, and cost efficiency (Newman, 2021). These systems consist of multiple interconnected nodes operating over a complex topology (Ashok et al., 2024; Hardt et al., 2024; Gu et al., 2024). To ensure the health of cloud systems, each node continuously monitors key performance indicators (KPIs), such as node latency, request counts, CPU utilization, and disk I/O (Meng et al., 2020). However, given the inherent complexity of these environments, faults are inevitable. These faults often manifest as anomalies—significant deviations from normal behavior—which are rare but can propagate across the system, impacting neighboring nodes and potentially compromising the entire application (Lomio et al., 2020). Timely identification of the *root cause* of such anomalies is crucial to minimizing downtime and reducing operational costs. While cloud systems can detect anomalies by identifying deviations in KPI patterns, automated root cause diagnosis (RCD) remains a significant challenge (Chen et al., 2019a).

We define root cause as nodes that satisfy two key conditions: (1) **Anomaly condition:** the root cause node exhibits anomalous behavior even while its causal parents are operating usually; and (2) **Fix condition:** had the root cause nodes assumed their usual values, the anomaly at the target node would not have occurred. While many prior RCD methods address the anomaly condition (Chen et al., 2014; Lin et al., 2018; Liu et al., 2021; Li et al., 2022), the fix condition is often overlooked, with some notable exceptions (Budhathoki et al., 2022b; Okati et al., 2024; Budhathoki et al., 2022a) that assess it via counterfactuals. Counterfactual estimation relies on learning a Structural Causal Model (SCM) (see Sec. A). Given a causal graph linking the system's nodes, an SCM learns a set of functional equations that model the generation of each node as a function of its causal parents

---

*Correspondence to: Lokesh Nagalapatti <nlokeshiisc@gmail.com>. Work done by L. N. and A. S. during an internship at Microsoft Research India.

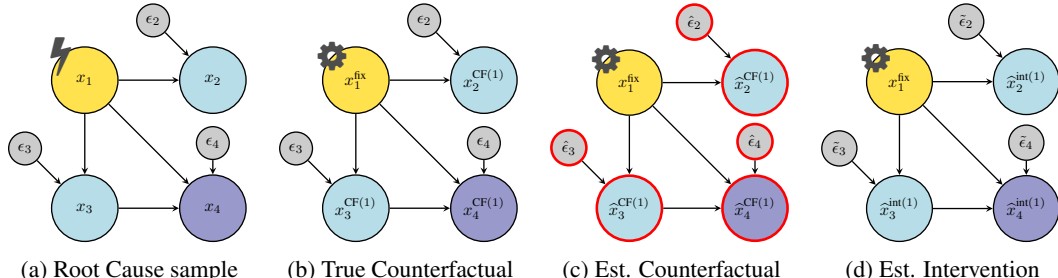

(a) Root Cause sample      (b) True Counterfactual      (c) Est. Counterfactual      (d) Est. Intervention

Figure 1: This figure demonstrates how abduction errors in counterfactuals impact RCD performance. In panel (a), an instance $x$ shows an anomaly at the purple target node $x_4$, with the root cause being the gold node $x_1$, which is affected by an abnormal intervention. Hence $x_1$ takes OOD values, and influences its downstream nodes also to take on OOD values. Latent exogenous nodes are shown in grey. Panel (b) illustrates the true CF $x^{\mathrm{CF}(1)}$ obtained by applying the fix. Panel (c) presents the estimated CF for the fix, using exogenous *estimates* $\hat{\epsilon}$ – involving abduction that requires SCM evaluations in OOD regions. Finally, panel (d) shows the interventional estimate, which uses *sampled* $\tilde{\epsilon}$, yielding a resulting $\hat{x}_4^{\mathrm{int}(1)}$ that conforms to the usual regime. Our theory in Sec 4 captures this rigorously.

and latent exogenous variables. Such causal graphs can be derived from inverted call graphs that are easily available for cloud deployments Hardt et al. (2024).

A key challenge with SCMs is that they are typically trained on historical data collected under usual operating conditions. As a result, while SCMs can produce accurate counterfactuals for in-distribution data, they become unreliable for anomalies that lie outside the training distribution (Okati et al., 2024). Our work reveals that these reliability issues stem from the first step in counterfactual estimation – abduction, where the functional equations in the SCM need to be inverted to estimate exogenous variables. Fig. 1 provides an illustration.

If the oracle values of latent exogenous variables were available, counterfactuals would be an ideal method for RCD. In practice, however, exogenous variables are never observed and must be estimated from causal mechanisms learned from observational training data. These causal mechanisms are trained on data predominantly representing usual behavior, leading to significant estimation errors in exogenous variables compared to their oracle values. To address this, we introduce IDI (**In-Distribution Interventions**), which evaluates the fix condition using in-distribution interventions. A key feature of IDI is that, when assessing the fix condition for the true root cause, it requires inference of the SCM only in in-distribution regions, ensuring a robust diagnosis. Our theoretical analysis compares IDI with counterfactual methods, demonstrating that the latter's error scales with the total variation distance between the usual training distribution and the rare anomalous distribution, which can be high for outliers. In contrast, IDI's error is bounded by the standard deviation of the latent exogenous variables. Experiments using cloud-based synthetic SCMs and a widely used RCD benchmark dataset with known causal graph (PetShop) demonstrate that IDI achieves greater robustness and accuracy in RCD compared to eleven baselines.

## 2 RELATED WORK

Several anomaly detection methods (Akoglu, 2021; Chandola et al., 2009) use an anomaly scoring function $g$ and a threshold $\tau$ to detect anomalies when an instance $x$ has $g(x) > \tau$. Our focus in this work is on RCD, and we group existing RCD approaches into Correlation and causal methods.

**Correlation Approaches.** Non-causal RCD methods typically rely on correlation-based analyses (Pham et al., 2024; cor, 2024; Chen et al., 2019b; Zhang et al., 1996; Chen et al., 2022; Luo et al., 2014; Ma et al., 2020; 2019; Yu et al., 2023) to assess relationships between a candidate root cause and the anomalous target node. Sometimes, spurious correlations make these methods predicting nodes that are not even causal ancestors of the target, leading to misleading conclusions.

**Causal Approaches.** Causal methods use a causal graph $\mathcal{G}$ to limit root cause search to ancestors. Some methods propose learning the graph $\hat{\mathcal{G}}$, and may yield better results than the Correlation ones. In cloud deployments, it is best to leverage the easily available call graphs as learning a causal graph from training dataset requires some strong, and untestable assumptions (Glymour et al., 2019).

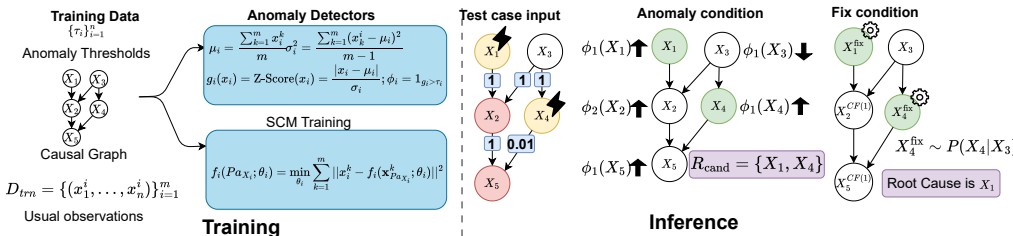

Figure 2: IDI's pipeline: During training, we use samples $D_{\text{trn}}$, a causal graph, and anomaly thresholds $\tau_i$ to learn anomaly detectors $\phi_i$, and structural equations $f_i$ as part of SCM training. During inference, we are given a root cause test case. The example illustrates two nodes $X_1, X_4$ with abnormal interventions. The edge weights indicate the strength of parent's influence (e.g., $\hat{f}_5(x_2, x_4) = x_2 + 0.01x_4$). While applying the anomaly condition, IDI discards $X_2$ in $\mathcal{R}_{\text{cand}}$ because of its anomalous parent $X_1$. Finally, the fix conditions excludes $X_4$ because it is insufficient to restore $X_5$ to its usual value. IDI declares $X_1$ as the root cause.

**Causal Anomaly Approaches.** Some methods approach RCD using just the anomaly condition to identify nodes with disrupted local causal mechanisms (Chen et al., 2014; Lin et al., 2018; Liu et al., 2021; He et al., 2022; Meng et al., 2020; Xin et al., 2023; Yang et al., 2022; Okati et al., 2024; Yu et al., 2021; Wang et al., 2018; Li et al., 2022; Shan et al., 2019). They predict nodes $x_j$ with low empirical sampling probabilities $P_{X_j}^{\text{trn}}(x_j|\text{Pa}_{x_j})$ conditioned on their causal parents as root causes. Some methods (Budhathoki et al., 2022a) attribute such abnormalities to latent exogenous disturbances, and they aim to infer abnormalities in the latent variables $\epsilon_j$. **Causal Fix Approaches.** Sometimes, an abnormal ancestor may have a negligible causal effect on the target, disqualifying it as a root cause. The fix condition becomes necessary to disregard such nodes. Prior methods assess the fix condition via soft interventions (Jaber et al., 2020; Ikram et al., 2022), interventions (Okati et al., 2024), and counterfactuals (Budhathoki et al., 2022b;a). SAGE (Gan et al., 2021) is a prominent method in this category that leverages conditional variational autoencoders to model SCMs and estimates counterfactuals by sampling from latent distribution predicted by the encoder. However, in the RCD context, the encoder itself is conditioned on OOD inputs during inference. Most methods probe their trained models with OOD inputs. In contrast, our approach, IDI, aims for RCD by probing its trained SCM on in-distribution inputs, thereby leading to a robust RCD approach.

## 3 PROBLEM FORMULATION

### 3.1 TRAINING SETUP

Our training dataset comprises the causal graph $\mathcal{G}$, and a set of $N$ samples $\{\boldsymbol{x}^i\}_{i=1}^N$, where each $\boldsymbol{x}^i \in \mathbb{R}^n$. These samples are assumed to be drawn from an oracle SCM, $\mathcal{S} = (\mathbf{X} = \{X_1, \ldots, X_n\}, \boldsymbol{\epsilon} = \{\epsilon_1, \ldots, \epsilon_n\}, \mathcal{F} = \{f_1, \ldots, f_n\}, P_{\boldsymbol{\epsilon}}^{\text{trn}})$. Our training dataset $D_{\text{trn}}$ encompasses samples that are obtained as follows from the Oracle SCM 1) sample $\boldsymbol{\epsilon}^i$ i.i.d. from $P_{\boldsymbol{\epsilon}}^{\text{trn}}$, and 2) compute $\boldsymbol{x}^i$ using the structural equations $\mathcal{F}$. Thus, most of the samples in $D_{\text{trn}}$ exhibit usual behavior. To assess the anomaly level of each node $x_i \in \boldsymbol{x}$, we train a set of anomaly detectors $\varphi = \{(g_1, \tau_1), (g_2, \tau_2), \ldots, (g_n, \tau_n)\}$. Each detector $\varphi_i$ consists of a scoring function $g_i : X_i \mapsto \mathbb{R}^+$ and an anomaly threshold $\tau_i > 0$. Thus, $\varphi_i(x_i) = 1_{g_i(x_i) > \tau_i}$ is the binary indicator for $x_i$ being anomalous. These detectors can be trained unsupervised on $D_{\text{trn}}$ using algorithms such as Z-score (Eidleman, 1995), isolation forest (Liu et al., 2008), or IT score (Budhathoki et al., 2022b). Given a test instance $\boldsymbol{x} = (x_1, \ldots, x_n)$ where an anomaly is detected at $x_n$, our **goal** is to trace the root cause of this anomaly to nodes in the system that caused it. We defer a detailed discussion on Structural Causal Models (SCMs) and the process of computing counterfactuals and interventions using SCMs to Appendix A.

### 3.2 QUALIFYING CRITERIA FOR ROOT CAUSE

For ease of exposition, let us assume there is a unique root cause $x_j$, and establish the criteria that $X_j$ needs to meet to qualify as a root cause. First, $X_j$ must be actionable; i.e., these must exist a fix value $x_j^{\text{fix}}$ such that applying it to $X_j$ avoids anomaly at $x_n$. Furthermore, the abnormality at the root cause node should not have originated from any of its ancestors; otherwise, resolving the anomaly

at $x_n$ may also require fixing other nodes upstream of $x_j$. More formally, a root cause should satisfy the following two criteria:

1. *Anomaly Condition:* $x_j$ must be anomalous, while its parent nodes are not, i.e., $\varphi_j(x_j) = 1$ and $\varphi_p(x_p) = 0$ for any parent node $p \in \mathrm{Pa}_{X_j}$.
2. *Fix Condition:* Setting $X_j$ to its fix value $x_j^{\mathrm{fix}}$ should have resolved the anomaly $x_n$. This implies the counterfactual $\boldsymbol{x}^{\mathrm{CF}(j)}$, obtained by intervening on $X_j = x_j^{\mathrm{fix}}$, should exhibit usual behavior at $X_n$; i.e., $\varphi_n(x_n^{\mathrm{CF}(j)}) = 0$.

**Defining the Root Cause Distribution.** We define the distribution $Q_X^{\mathrm{RC}(j)}$ that governs unique root cause samples $\boldsymbol{x}$, where $x_j$ is the root cause of anomaly at $x_n$. Such a distribution will have $\boldsymbol{\epsilon}_{-j}$, representing all exogenous variables except $\epsilon_j$, drawn from their usual distribution. Whereas, $\epsilon_j$ must be sampled strong enough to induce an anomaly at both $X_n, X_j$.

**Definition 1** *We define the anomalous distribution $Q_X^{RC(j)}$ for unique root cause at $X_j$ as:*

$$Q_{\boldsymbol{\epsilon}}^{RC(j)}(\boldsymbol{\epsilon}) = Q_{\boldsymbol{\epsilon}}^{RC(j)}(\boldsymbol{\epsilon}_{-j})Q_{\boldsymbol{\epsilon}}^{RC(j)}(\epsilon_j|\boldsymbol{\epsilon}_{-j}) \tag{1}$$

*These two factors are defined as:*

$$Q_{\boldsymbol{\epsilon}}^{RC(j)}(\boldsymbol{\epsilon}_{-j}) = P_{\boldsymbol{\epsilon}}^{trn}(\epsilon_{-j}) \tag{2}$$

$$Q_{\boldsymbol{\epsilon}}^{RC(j)}(\epsilon_j|\boldsymbol{\epsilon}_{-j}) = P_{\boldsymbol{\epsilon}}^{trn}\left(\epsilon_j|\boldsymbol{\epsilon}_{-j}, \varphi_j(x_j) = 1, \varphi_n(x_n) = 1\right) \tag{3}$$

*We denote the distribution induced by $Q_{\boldsymbol{\epsilon}}^{RC(j)}$ on the observed $X$ through the SCM $\mathcal{S}$ as $Q_X^{RC(j)}$.*

We determine the fix value $x_j^{\mathrm{fix}}$ to be applied to the root cause node by sampling from the conditional distribution $P_X^{\mathrm{trn}}(X_j \mid \mathrm{Pa}_{x_j})$, reflecting how humans would naturally adjust $X_j$ in practice. This approach also aligns with prior research (Budhathoki et al., 2022b), which samples an exogenous fix $\epsilon_j^{\mathrm{fix}}$ to induce usual values in $X_j$ as a consequence. To estimate the causal effect of propagating $x_j^{\mathrm{fix}}$ downstream to the target node $X_n$, we consider two approaches: the counterfactual estimate $\widehat{x}_n^{\mathrm{CF}(j)}$ from Budhathoki et al. (2022b) and an alternative interventional estimate $\widehat{x}_n^{\mathrm{int}(j)}$ that we propose. We begin with a theoretical analysis to compare the errors associated with these two methods.

## 4 Interventions vs. Counterfactuals for RCD

We begin with the definitions used in our theoretical analysis.

**Definition 2** *For two distributions $P, Q$ defined over a space $\mathcal{X}$, the total variational distance (Redko et al., 2019) between them is defined as: $\mathrm{tvd}(P, Q) = \frac{1}{2}\int_{x \in \mathcal{X}} |P(x) - Q(x)|dx$.*

**Definition 3** *We say that an SCM $\mathcal{S}$ is an additive noise model when the structural equations are of the form $x_i = f_i(\mathrm{Pa}_{X_i}) + \epsilon_i$, where $f_i$ is a deterministic function, and $\epsilon_i$ is the exogenous variable.*

Let $\hat{P}_{\epsilon_i}^{\mathrm{trn}}$ denote the estimate for latent exogenous distribution $P_{\epsilon_i}^{\mathrm{trn}}$, obtained from a validation dataset $D_V$. For additive noise models, $\hat{P}_{\epsilon_i}^{\mathrm{trn}}$ is simply the empirical distribution of the error residuals $x_i - \hat{f}_i(\mathrm{Pa}_{x_i})$ computed for each $\boldsymbol{x} \in D_V$. We need to use validation set to avoid the overfitting bias in estimation, that otherwise arises from using the train set (Chernozhukov et al., 2018).

**Definition 4** *Consider an additive noise model $\mathcal{S}$, and let $\widehat{\mathcal{S}}$ be its estimate learned from training data $(D_{trn}, \mathcal{G})$. For a fix $x_j^{fix}$ applied to the node $X_j$, let $\boldsymbol{x}^{CF(j)}$ be the true counterfactual from $\mathcal{S}$, and $\widehat{\boldsymbol{x}}^{CF(j)}, \widehat{\boldsymbol{x}}^{int(j)}$ be the estimated counterfactual and intervention from $\widehat{\mathcal{S}}$. Then, the following equations show how these quantities are derived:*

$$x_{j+1}^{CF(j)} = f_{j+1}(x_j^{fix}) + \epsilon_{j+1} \text{ where } \epsilon_{j+1} = x_{j+1} - f_{j+1}(x_j) \tag{4}$$

$$\widehat{x}_{j+1}^{CF(j)} = \hat{f}_{j+1}(x_j^{fix}) + \hat{\epsilon}_{j+1} \text{ where } \hat{\epsilon}_{j+1} = x_{j+1} - \hat{f}_{j+1}(x_j) \tag{5}$$

$$\widehat{x}_{j+1}^{int(j)} = \hat{f}_{j+1}(x_j^{fix}) + \tilde{\epsilon}_{j+1} \text{ where } \tilde{\epsilon}_{j+1} \sim \hat{P}_{\epsilon_{j+1}} \tag{6}$$

*This procedure iterates for $j + 2, \ldots, n$ in a topological order.*

Since assessing the fix relies on $\varphi_n(x_n^{\mathrm{CF}(j)})$, we bound the error incurred in estimating the true $x_n^{\mathrm{CF}(j)}$ derived from the Oracle SCM $\mathcal{S}$ using $\widehat{x}_n^{\mathrm{CF}(j)}$ from a fitted SCM $\widehat{\mathcal{S}}$.

**Theorem 5** *Suppose the oracle SCM $\mathcal{S}$ is an additive noise model over a chain graph $\mathcal{G} = X_1 \to \cdots \to X_n$, with structural equations of the form $f_i(x_{i-1}) + \epsilon_i$, where each $\epsilon_i$ has bounded variance $\sigma^2$, and each function $f_i$ is $K$-Lipschitz. Consider the hypothesis class $\mathcal{H} = \{\mathcal{H}_i\}_{i=1}^n$, where each $\mathcal{H}_i$ comprises bounded $K$-Lipschitz functions, resulting in losses bouned by $M > 0$. Let $\widehat{\mathcal{S}}$ be the SCM fitted on training data $D_{trn}$, with estimated functions $\{\hat{f}_i\}_{i=1}^n$. Then, for test samples $\boldsymbol{x}$ drawn from the root cause distribution $Q_X^{RC(j)}$, with $X_j$ as the unique root cause; for a fix $x_j^{fix} \sim \hat{P_X^{trn}}(X_j|x_{j-1})$ sampled from its empirical distribution, the estimated counterfactual $\widehat{x}_n^{CF(j)}$ computed from $\widehat{\mathcal{S}}$ satisfies:*

$$
\mathbb{E}_{\boldsymbol{x} \sim Q_X^{RC(j)}}[|x_n^{CF(j)} - \widehat{x}_n^{CF(j)}|] \leq \sum_{i>j} K^{n-i} \Bigg[ 2^{n-i+1} \mathbb{E}_{x_{i-1} \sim P_{X_{i-1}}^{trn}} \left[ |f_i(x_{i-1}) - \hat{f}_i(x_{i-1})| \right]
$$
$$
+ M^{n-i+1} \cdot \left( tvd\big(P_{X_{i-1}}^{trn}, Q_{X_{i-1}}^{RC(j)}\big) + tvd\big(P_{X_{i-1}}^{trn}, P_{X_{i-1}}^{\hat{trn}}\big) \right) \Bigg]
\tag{7}
$$

*Proof Sketch:* The main issue stems from abduction in Eq. 12, where $\hat{f}_{j+1}$ is inferred at an OOD input $x_j$. This introduces $tvd(P, Q)$ terms in the bound. The exogenous error at a node propagates downstream to their descendants, further compounding the error at the target node. Please refer Fig. 1 for an illustration.

**Remark:** We first discuss the $tvd(P_{X_{i-1}}^{trn}, Q_{X_{i-1}}^{RC(j)})$ terms that arise from *abduction*. In a geometric series, the leading term dominates the sum. Here, the leading term involves $tvd(P_{X_j}^{trn}, Q_{X_j}^{RC(j)})$, where $\varphi_j(x_j) = 1$. Thus, for the root cause $X_j$ the above discrepancy lies between the usual training distribution and the anomalous test distribution. Since this discrepancy is substantial, $\widehat{x}_n^{\mathrm{CF}(j)}$ is a poor estimate of $x_n^{\mathrm{CF}(j)}$. Other terms in the bound reflect discrepancies between i.i.d. train, test errors, and can be shown to reduce with increasing training size $|D_{\mathrm{trn}}|$ using generalization bounds like VC-dimensions, etc. (Redko et al., 2019).

Next, we conduct a similar analysis to assess the error in using $\widehat{x}_n^{\mathrm{int}(j)}$ as an estimate for $x_n^{\mathrm{CF}(j)}$.

**Theorem 6** *Under the same conditions laid out in Theorem 5, the error between the true counterfactual $x_n^{CF(j)}$ and the estimated intervention $\widehat{x}_n^{int(j)}$ admits the following bound:*

$$
\mathbb{E}_{\boldsymbol{x} \sim Q_X^{RC(j)}} \left[ |x_n^{CF(j)} - \widehat{x}_n^{int(j)}| \right] \leq \sum_{i>j} K^{n-i} \Bigg[ \mathbb{E}_{x_{i-1} \sim P_{X_{i-1}}^{trn}} \left[ |f_i(x_{i-1}^{CF(j)}) - \hat{f}_i(x_{i-1}^{CF(j)})| \right]
$$
$$
+ M^{n-i+1} \, \mathrm{tvd}\left( P_{X_{i-1}}^{trn}, \hat{P^{trn}}_{X_{i-1}} \right) + \mathrm{std}(\epsilon_i)
$$
$$
+ \mathrm{std}(\tilde{\epsilon}_i) + \big| \mathbb{E}[\epsilon_i] - \mathbb{E}[\tilde{\epsilon}_i] \big| \Bigg]
\tag{8}
$$

*where std(●) denotes the standard deviation.*

*Proof Sketch:* Unlike in the CF case, the estimation of $\widehat{x}_n^{\mathrm{int}(j)}$ does not need abduction. Instead, it samples $\tilde{\epsilon}_{j+1}$ as shown in Eq. 6, and the error from abduction is limited by the standard deviation of the oracle $\epsilon_j$. A key advantage of this is that the challenging $tvd(P, Q)$ terms are eliminated from the bound, and these samples $\tilde{\epsilon}_{j+1}$ result in only in-distribution inputs to the fitted SCM.

**Corollary 7** *Suppose in Theorem 6, the exogenous variables $\epsilon_i$ are zero-mean in addition to having bounded variance for any $i \in [n]$, then the two terms $std(\tilde{\epsilon}_i)$ and $|\mathbb{E}[\epsilon_i] - \mathbb{E}[\tilde{\epsilon}_i]|$ can be dropped from the bound in Theorem 6.*

**Remark:** All terms, except $\text{std}(\epsilon_i)$, can be simplified using generalization bounds. Importantly, interventional estimates remain stable despite the drift between the training distribution $P_X^{\text{trn}}$ and the anomalous distribution $Q_X^{\text{RC}(j)}$. Thus, when exogenous variables have low variance, interventions provide a more robust method for estimating $x_n^{\text{CF}(j)}$ and evaluating the fix condition.

We defer all proofs to Appendix C.

## 5 OUR APPROACH: IDI

IDI uses interventions motivated by the analysis in Sec. 4 as they offer a robust approach to RCD. We illustrate the training and inference procedure of IDI in Fig. 2. IDI first applies the anomaly condition to filter the promising root cause candidates into a set $\mathcal{R}_{\text{cand}}$. Then, it applies the fix condition to nodes in $\mathcal{R}_{\text{cand}}$ to diagnose the root cause. We lay down the procedure for unique root cause and multiple root cause scenarios separately. For multiple root cause diagnosis, we need the following assumption to be met:

**Assumption 1** *At most one root cause exists in every simple path that leads to the target $X_n$ in $\mathcal{G}$.*

**Step 1 of IDI: Assessing the anomaly condition.** The goal of this step is to identify nodes whose exogenous variables are abnormal. IDI's approach is straightforward; it iterates over all ancestors of $X_n$ in $\mathcal{G}$, adding node $X_i$ to $\mathcal{R}_{\text{cand}}$ if $X_i$ shows an anomaly but none of its parents do. Theorem 4.5 in (Okati et al., 2024) proves that this approach is sound for chain graphs. To assess anomalies, we use the Z-Score,

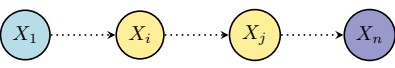

Figure 3: A chain graph with more than one root cause in a simple path. Dotted lines denote a directed path.

defined for $X_i$ as Z-score$(x_i) = \frac{|x_i - \mu_i|}{\sigma_i}$ where $\mu_i$ and $\sigma_i$ are the sample mean and standard deviation computed for the $i^{\text{th}}$ node in the training data. IDI can easily accommodate any other methods proposed for the anomaly criterion (Chen et al., 2014; Li et al., 2022; Okati et al., 2024).

Next, We illustrate the need for assumption 1 using a chain graph in Fig. 3 that comprises two root cause nodes, $X_i$ and $X_j$, along a simple directed path to $X_n$. Since $X_i$ is a root cause, it is anomalous and is likely to influence its downstream nodes to take on anomalous values, including $X_j$'s parent. As a result, IDI may discard $X_j$ incorrectly.

**Step 2 of IDI: Assessing the fix Condition.** We first describe the procedure for the simpler of unique root cause, and then generalize to multiple root causes.

*Unique root cause:* In this case, IDI applies the fix condition on nodes in $\mathcal{R}_{\text{cand}}$ iteratively. Suppose $X_j$ is the true cause, our fix value $x_j^{\text{fix}}$ applied to $X_j$ suppresses the only abnormal $\epsilon_j$ that caused the anomaly $x_n$. Since IDI samples all other exogenous variables downstream of $X_j$ from their usual distributions, the fix condition for true root cause is assessed by probing the learned SCM at in-distribution inputs. Therefore, $\varphi_n(\widehat{x}_n^{\text{int}(j)})$ would evaluate to 0 and IDI correctly predicts $X_j$.

*Multiple root causes:* In this case, we require set-valued fixes to avoid OOD evaluations. We justify this need with an example in Fig. 4. Let $\alpha^\star = \{X_i, X_j\}$ be two root cause nodes. Since they intersect at a common descendant $X_k$, a fix applied to $X_i$ leads to an OOD evaluation at $X_k$ due to the influence of its anomalous ancestor $X_j$. This can potentially causing errors in $\widehat{x}_k^{\text{int}(i)}$ inference that propagate to $\widehat{x}_n^{\text{int}(i)}$. However, when both $X_i$ and $X_j$ are fixed simultaneously, all evaluations will be in-distribution.

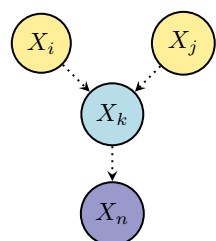

Figure 4: An example graph with more than one root cause.

Another issue with interventions is that any superset of $\alpha^\star$ may appear to resolve the anomaly, leading to over-prediction of root causes. For instance, applying a fix to $\alpha \subset \alpha^\star$ suppresses the abnormalities in both the root causes in $\alpha^\star$ and also some redundant nodes in $\alpha^\star \setminus \alpha$. To address this, we leverage Shapley values (Shapley, 1953) from cooperative game theory. Given a set $\mathcal{U}$ of $m$ items and a utility function $\phi : 2^{\mathcal{U}} \mapsto \mathbb{R}^+$ that assigns a score to each subset $\alpha \subseteq \mathcal{U}$, Shapley values provide a fair method to distribute credit among the $m$ players. A player $i$ that achieves a high score of $\phi(\alpha \cup \{i\}) - \phi(\alpha)$ across multiple $\alpha \subset \mathcal{U}$ receives a high Shapley value. We treat $\mathcal{R}_{\text{cand}}$ as the items and define the utility function for $\alpha \subseteq \mathcal{R}_{\text{cand}}$ as

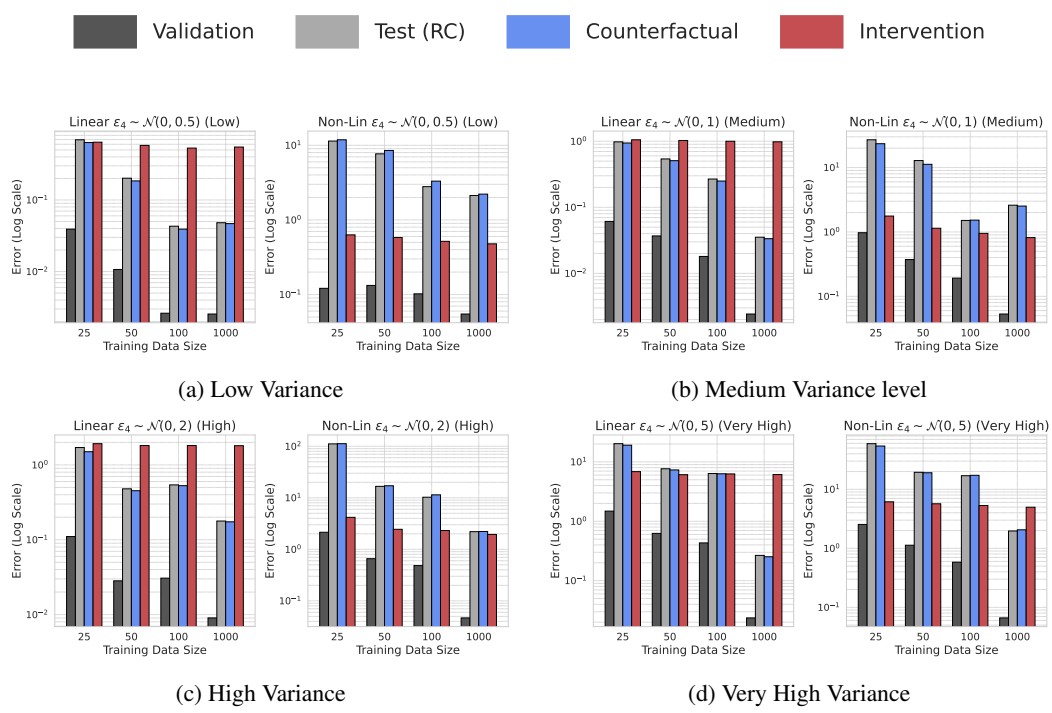

Figure 5: Assessing the impact of variance of $\epsilon_i$ using a four variable additive noise toy dataset.

the reduction in raw anomaly scores at $X_n$ after fixing $\alpha$: $g_n(x_n) - g_n(\widehat{x}_n^{\text{int}(\alpha)})$. Any subset that includes $\alpha^\star$ reduces the raw anomaly scores significantly thereby achieving high Shapley values. The pseudocode for the multi-root-cause diagnosis algorithm is presented in Alg 1 in the Appendix.

# 6 EXPERIMENTS

We conduct a toy experiment on a four-variable dataset to showcase when interventions outperform counterfactuals and vice versa. We then test our approach on PetShop and multiple synthetic datasets, comparing it against a broader set of baseline RCD methods.

## 6.1 TOY EXPERIMENTS

We consider a causal graph where the root nodes $X_1, X_2, X_3$ each have an edge to a common child $X_4$. We instantiate two Oracle SCMs: a linear additive noise model and a non-linear one. We sample the linear weights $w_1, w_2, w_3$ from $\mathcal{N}(0, 1)$ and define the non-linear model as $X_4 = \frac{\sin(X_1) + \sqrt{|X_2|} + \exp(-X_3)}{3} + \epsilon_4$. We draw the exogenous variables $\epsilon_1, \epsilon_2, \epsilon_3$ from $\mathcal{N}(0, 1)$ and define the structural equations for the root nodes as $X_i = f_i(\epsilon_i) = \epsilon_i$ for $i \in \{1, 2, 3\}$. To create root cause test samples, we assign one of $\{X_1, X_2, X_3\}$ a value from $U[3, 10]$, ensuring at least a 3-standard deviation shift that induces an abnormality at $X_4$. We generate $n \in \{25, 50, 100, 1000\}$ training samples, along with 100 validation and 100 test samples, each with a unique root cause. Since we consider a simple SCM with a single function over 3 variables and the linear model is estimated in closed form, small training sample sizes are chosen to study SCM fitting errors. Learning $\widehat{S}$ reduces to fitting $\hat{f}_4$: we fit the linear model using closed-form regression and train the non-linear model as a three-layer MLP with 10 hidden nodes and ReLU activations via gradient descent. For the toy experiment, we sample $x_j^{\text{fix}}$ from its *true* distribution $\mathcal{N}(0, 1)$. We assess the following errors:

1. Validation Error: Measures the accuracy of $\hat{f}_4$ on in-distribution data; $|f_4(\boldsymbol{x}_{1:3}^{\text{val}}) - \hat{f}_4(\boldsymbol{x}_{1:3}^{\text{val}})|^2$.
2. Test (RC) Error: Measures the prediction accuracy on OOD samples; $|f_4(\boldsymbol{x}_{1:3}^{\text{RC}}) - \hat{f}_4(\boldsymbol{x}_{1:3}^{\text{RC}})|^2$
3. Counterfactual Error: Quantifies the error arising from using CFs; $|x_4^{\text{CF}(j)} - \widehat{x}_4^{\text{CF}(j)}|^2$.
4. Interventional Error: Measures the error for using interventions; $|x_4^{\text{CF}(j)} - \widehat{x}_4^{\text{int}(j)}|^2$.

We defer a detailed description of the toy setup and additional experiments to Appendix D. Figure 5 presents results across different $\epsilon_4$ variance levels. Each panel in the figure compares linear (left) and non-linear (right) settings. We make the following key observations.

- **Low Variance:** Linear models generalize well to the OOD root cause distribution, leading to small CF errors (Fig. 5a). Non-linear models, however, overfit, resulting in high CF errors.
- **Medium Variance:** Trends remain consistent, confirming that CF error correlates with OOD test error whereas interventional error remains bounded by the standard deviation of $\epsilon_4$.
- **High Variance:** Interventions outperform CFs, especially in low-data regimes, as high $\epsilon_4$ variance destabilizes the learning $\hat{f}_4$ leading to very poor OOD generalization.
- **Very High Variance:** CFs slightly outperform interventions, but only because the additive noise assumption holds. This fails when the assumption is violated, as we will see in Sec. 6.4 (RQ3).

**Summary:** This simple toy experiment shows that while interventional error is bounded by the standard deviation of $\epsilon_4$, CF error closely aligns with the test error on OOD root cause samples. Thus, when the learned SCM deviates from the true SCM on distributions unseen during training, interventional estimates provide a more reliable approach to root cause diagnosis.

## 6.2 COMPARISON WITH BASELINES

We compare IDI against following baselines:

- **Correlation** This class includes `ε-diagnosis` (Shan et al., 2019), which identifies root causes by testing for significant behavior changes during anomalies, and `ranked correlation` (Hardt et al., 2024), which assigns root cause to nodes correlated with the target.
- **Causal Anomaly** This class encompasses `traversal` methods (Chen et al., 2014; Lin et al., 2018; Liu et al., 2021; Meng et al., 2020), broadly grouped by (Okati et al., 2024), including the `smooth traversal` method introduced by them. These methods implement a version our anomaly condition. `CIRCA` (Li et al., 2022) identifies nodes connected to the target through fully anomalous paths, while `random walk` methods (Yu et al., 2021) use heuristics.
- **Causal Fix** This includes Hierarchical RCD (`HRCD`) (Ikram et al., 2022), which predicts root causes as nodes that suffered local mechanism changes affecting the target. `TOCA` (Okati et al., 2024) implements the fix condition jointly over all nodes. The `CF Attribution` (Budhathoki et al., 2022b) method uses Shapley values to perform CF contribution analysis on *all* $n$ nodes.

We implemented IDI in the RCD library released by PetShop (Hardt et al., 2024)[1]. PetShop uses Dowhy (Sharma & Kiciman, 2020) and gcm (Blöbaum et al., 2022) for causal inference and PyRCA (Liu et al., 2023) for root cause analysis. To ensure fair comparisons, we applied the same experimental settings for IDI as those used for the baselines in PetShop.

**Evaluation Metric.** We assess root cause prediction accuracy using Recall@k (Ikram et al., 2022). Let $\alpha^\star \subset \mathbf{X}$ be the ground truth root causes, and $\hat{\alpha}$ the ordered list of predicted root causes, where $\hat{\alpha}[1]$ is the most prominent. For any $k > 0$

$$\text{Recall@k}(\hat{\alpha}, \alpha^\star) = \frac{\sum_{i=1}^{|\alpha^\star|+k-1} 1_{\hat{\alpha}[i] \in \alpha^\star}}{|\alpha^\star|}; \quad 1_{\hat{\alpha}[i] \in \alpha^\star} = 1 \text{ if } \hat{\alpha}[i] \in \alpha^\star \tag{9}$$

For $k = 1$, we see that $\text{Recall@1}(\hat{\alpha}, \alpha^\star) = 1$ iff every node in $\alpha^\star$ is present in $\hat{\alpha}[1:k]$, while for larger $k$, we need $\alpha^\star$ to be present in the first $|\alpha^\star|+k-1$ predictions of $\hat{\alpha}$. We assess using $k = 1, 3$.

## 6.3 EXPERIMENTS ON PETSHOP: RCD IN A DEPLOYED CLOUD SYSTEM

PetShop (Hardt et al., 2024) is a recent dataset designed for benchmarking RCD methods in the cloud domain, featuring a call graph $\mathcal{G}$ that causally links key performance indicators (KPIs). The baseline methods in the PetShop library use a linear additive noise model $\hat{\mathcal{S}}$, which we also adopted for IDI. This dataset encompasses three types of latency issues: low, high, and temporal, with many methods successfully identifying the temporal issues. Overall, IDI outperforms other methods in most settings, except for Recall@3 in high latency, where `CIRCA` emerged as the best performer. `CIRCA` identifies root causes as nodes connected to the target through all anomalous nodes, and

---

[1] https://github.com/amazon-science/petshop-root-cause-analysis

| | | Low | | High | | Temporal | |
|---|---|---|---|---|---|---|---|
| Recall@ | | k=1 | k=3 | k=1 | k=3 | k=1 | k=3 |
| Correlation | Random Walk (Yu et al., 2021) | 0.00 | 0.10 | 0.00 | 0.20 | 0.00 | 0.33 |
| | Ranked Correlation (Hardt et al., 2024) | 0.40 | 0.60 | 0.70 | 0.90 | 0.50 | 0.67 |
| | $\epsilon$-Diagnosis (Shan et al., 2019) | 0.00 | 0.00 | 0.00 | 0.00 | 0.17 | 0.17 |
| Causal Anomaly | Circa (Li et al., 2022) | 0.60 | 0.80 | 0.60 | **1.00** | 0.67 | **1.00** |
| | Traversal (Chen et al., 2014) | 0.80 | 0.80 | **0.90** | 0.90 | **1.00** | **1.00** |
| | Smooth Traversal (Okati et al., 2024) | 0.40 | 0.60 | 0.00 | 0.60 | 0.50 | **1.00** |
| Causal Fix | HRCD (Ikram et al., 2022) | 0.07 | 0.21 | 0.00 | 0.07 | 0.25 | 0.75 |
| | TOCA (Okati et al., 2024) | 0.40 | 0.40 | 0.20 | 0.20 | 0.00 | 0.00 |
| | CF Attribution (Budhathoki et al., 2022b) | 0.40 | 0.60 | 0.40 | 0.70 | 0.00 | 0.50 |
| | IDI (Ours) | **0.90** | **0.90** | **0.90** | 0.90 | **1.00** | **1.00** |

Table 1: Diagnosing root causes of latency issues in PetShop, a cloud-based microservices dataset. Methods are categorized into Correlation, Causal Anomaly, and Causal Fix.

possibly high latency test cases showed such a favorable behavior. Otherwise, `Traversal` proved to be a strong contender against IDI. The improvements observed in IDI across various settings can be attributed solely to its robust implementation of the fix condition. In contrast, the gains seen in other methods assessing the fix condition are less pronounced compared to Causal Anomaly methods, as their evaluations involve applying $\widehat{\mathcal{S}}$ to out-of-distribution inputs.

## 6.4 EXPERIMENTS ON SYNTHETIC SCMS

Next, we design synthetic experiments to answer three key research questions:

RQ1 *Linear SCM:* How effective is RCD when the Oracle SCM $\mathcal{S}$ has linear structural equations, so that an $\hat{\mathcal{S}}$ fit using samples from the usual distribution also generalizes OOD?

RQ2 *Non-Lin Invertible SCM:* How effective is RCD when $\hat{\mathcal{S}}$ closely approximates a non-linear $\mathcal{S}$ within the usual regime, and $\mathcal{S}$ allows abduction, meaning $f_i$ is invertible with respect to $\epsilon_i$?

RQ3 *Non-Lin Non-Invertible SCM:* What are the implications for root cause identification when $\hat{\mathcal{S}}$ closely matches a non-linear $\mathcal{S}$ in the usual distribution, but $\mathcal{S}$ does not support abduction?

We evaluate each option under unique and multiple root cause scenarios. For multiple root causes, we ensure they follow the assumption 1, and later perform ablations under its violations in Table 5. Our synthetic setup involved a single anomalous test sample for RCD, so we did not run two baselines: 1) $\epsilon$-diagnosis method, which requires multiple anomalous samples for conducting two-sample tests, and 2) HRCD which learns the causal graph solely from anomalous samples.

**Generating the Oracle SCM $\mathcal{S}$:** We randomly sample a causal graph $\mathcal{G}$ using the Networkx library Hagberg et al. (2008). We select the node with the most ancestors as the anomaly target, and the root cause nodes $\alpha^\star$ arbitrarily. Since cloud KPIs, such as node latency and CPU utilization, are typically positive (Meng et al., 2020), we ensured that all nodes in the synthetic data assume positive values. Exogenous variables follow a uniform distribution $\epsilon_i \sim U[0,1]$ making their standard deviation $\text{std}(\epsilon_i) = 0.3$. We explore other choices for $\epsilon_i$ in Appendix G. Finally, We generate the training dataset $D_{\text{trn}}$ by sampling exogenous variables $\boldsymbol{\epsilon}$, and then use $\mathcal{S}$ to generate the observed nodes $\boldsymbol{x}$ in a topological order. Each node in $\mathcal{G}$ is assigned a local causal mechanism as follows:

- For linear SCMs in RQ1, we define the functional equations as linear, with random coefficients as: $x_i = w_i^\top \text{Pa}_{x_i}$ where $w_i \sim U[0.5, 2]$.
- For non-lin invertible SCMs in RQ2, we set each local mechanism $f_i$ as an additive noise three layer ELU activation based MLP.
- For non-lin non-invertible SCMs, we use the same MLP architecture, but without additive noise. In this case, each MLP $f_i$ receives both the parents $\text{Pa}_{x_i}$ and the noise $\epsilon_i$ as inputs.

For linear SCM, we fit a linear additive noise SCM $\widehat{\mathcal{S}}$, whereas for both RQ2 and RQ3, we fit an additive noise MLP-based SCM $\widehat{\mathcal{S}}$. We show some example graphs in Appendix I.

**Test case generation:** For the root cause set $\alpha^\star$ to cause an anomaly at $x_n$, we first sample $\boldsymbol{\epsilon}_{-\alpha^\star}$ from $U[0,1]$, then apply a grid search over $\boldsymbol{\epsilon}_{\alpha^\star}$ so that $(\boldsymbol{\epsilon}_{\alpha^\star}, \boldsymbol{\epsilon}_{-\alpha^\star})$ together lead the SCM $\mathcal{S}$ to induce an anomaly at $x_n$. We also introduced some irrelevant anomalies at nodes that have weak functional relationships with $x_n$ to assess the impact on methods that ignore the fix condition. We repeat each experiment 10 times and report the average values of Recall@$k \in \{1, 3\}$.

| Number root causes | RQ1 Linear | | | | RQ2 Non-Lin Invertible | | | | RQ3 Non-Lin Non-Invertible | | | |
|---|---|---|---|---|---|---|---|---|---|---|---|---|
| | Unique | | Multiple | | Unique | | Multiple | | Unique | | Multiple | |
| Recall@ | $k=1$ | $k=3$ | $k=1$ | $k=3$ | $k=1$ | $k=3$ | $k=1$ | $k=3$ | $k=1$ | $k=3$ | $k=1$ | $k=3$ |
| Random Walk | 0.00 | 0.00 | 0.03 | 0.03 | 0.10 | 0.30 | 0.27 | 0.27 | 0.10 | 0.30 | 0.27 | 0.27 |
| Ranked Correlation | 0.00 | 0.00 | 0.00 | 0.00 | 0.00 | 0.00 | 0.00 | 0.00 | 0.00 | 0.00 | 0.00 | 0.00 |
| Traversal | 0.50 | **1.00** | 0.40 | 0.67 | 0.00 | **1.00** | 0.03 | 0.48 | 0.00 | **0.80** | 0.00 | 0.33 |
| Smooth Traversal | 0.50 | **1.00** | 0.52 | **0.90** | 0.50 | **1.00** | **0.57** | **0.83** | 0.40 | **0.80** | 0.40 | 0.73 |
| CIRCA | 0.80 | **1.00** | 0.33 | 0.33 | 0.60 | 0.80 | 0.38 | 0.48 | 0.40 | 0.70 | 0.03 | 0.03 |
| TOCA | 0.00 | 0.00 | 0.00 | 0.00 | 0.00 | 0.00 | 0.00 | 0.00 | 0.00 | 0.00 | 0.00 | 0.00 |
| CF Attribution | **1.00** | **1.00** | 0.97 | **1.00** | 0.30 | 0.70 | 0.20 | 0.40 | 0.00 | 0.20 | 0.00 | 0.23 |
| IDI (ours) | **1.00** | **1.00** | **1.00** | **1.00** | **1.00** | **1.00** | **0.83** | **0.97** | **0.60** | **0.80** | **0.63** | **0.83** |

Table 2: Recall@k=1 and 3 values for experiments on synthetic SCMs. We present the recall values averaged across ten runs. Best methods are highlighted in green, while the second best methods are shown in yellow. Overall, IDI demonstrates the highest recall across all settings.

**Results:** Table 2 summarizes our findings. Correlation methods struggle across all settings as root cause nodes correlate all descendants with the target $X_n$. In linear SCMs, many methods detect the unique root cause. However, with multiple root causes, the number of anomalous downstream nodes increases significantly, leading to more false positives. Among Causal Anomaly methods, `smooth traversal` performs best. `CF Attribution` excels *only* in linear SCMs due to accurate abduction. `TOCA` evaluates all nodes indiscriminately, causing numerous OOD SCM evaluations, while IDI efficiently focuses on $\mathcal{R}_{\mathrm{cand}}$. Overall, IDI achieves the best performance.

## 6.5 EXPERIMENT ON IDI VS. IDI (USING CFs)

| | Synthetic Oracle SCM | | | | | | PetShop Latency | | |
|---|---|---|---|---|---|---|---|---|---|
| | Linear | | Non-Lin Inv | | Non-Lin Non Inv | | low | high | temporal |
| | Unique | Multiple | Unique | Multiple | Unique | Multiple | | | |
| CF Attribution | **1.00** | 0.97 | 0.30 | 0.20 | 0.00 | 0.00 | 0.40 | 0.40 | 0.00 |
| IDI (CF) | **1.00** | **1.00** | **1.00** | 0.77 | 0.60 | 0.12 | 0.70 | 0.70 | 1.00 |
| IDI | **1.00** | **1.00** | **1.00** | **0.83** | **0.70** | **0.63** | **0.90** | **0.90** | **1.00** |

Table 3: Recall@1 for CF Attribution - as baseline method (Budhathoki et al., 2022b), IDI (CF) a version of IDI that uses CF in step-2, and IDI. IDI performs the best.

In this experiment, we run IDI in counterfactual (CF) mode, denoted as IDI (CF). IDI (CF) first filters root cause candidates $\mathcal{R}_{\mathrm{cand}}$ and then applies Shapley analysis on estimated counterfactuals instead of interventions. As a baseline, we compare a CF attribution method that skips $\mathcal{R}_{\mathrm{cand}}$ filtering and applies CF Shapley analysis to all nodes. Table 3 shows that CF attribution underperforms IDI (CF), confirming that filtering $\mathcal{R}_{\mathrm{cand}}$ isolates promising candidates for in-distribution SCM evaluations. While IDI (CF) excels in linear settings, its performance drops elsewhere compared to IDI, solely due to errors in abduction, highlighting interventions as the more effective approach.

## 7 LIMITATIONS AND CONCLUSION

**Limitations:** (1) IDI's performance can degrade when Assumption 1 is violated. (2) In additive noise models, when the training size is large enough for the estimated SCM to approach the oracle SCM, errors in using interventional estimates of counterfactuals plateau at the std. deviation of the exogenous variables (Appendix G). Whereas, counterfactual estimates converge to the true values.

**Conclusion:** In this paper, we introduced two key conditions—Anomaly and Fix—to identify root causes of anomalies. While prior methods effectively addressed the anomaly condition, the fix condition often relied on probing trained models with OOD inputs. To address this, we proposed IDI, a novel in-distribution intervention method that ensures the fitted SCM is probed using in-distribution inputs while evaluating the potential of true root cause nodes to resolve the anomaly. Unlike previous methods that required a unique root cause assumption, IDI operates under the more relaxed condition of at most one cause per path. We showed theoretically that IDI's intervention method is superior to counterfactual approaches for additive noise chain SCMs. Our experiments with arbitrary SCMs reaffirmed IDI's capability to deliver robust and accurate RCD.

## 8 ACKNOWLEDGEMENTS

We thank Avishek Ghosh and Abhishek Pathapati for their help in reviewing the proofs. We also thank Srinivasan Iyengar and Shivkumar Kalyanaraman from Microsoft for providing motivating real-world examples and feedback on our solutions.

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

## A   PRELIMINARIES ON STRUCTURAL CAUSAL MODELS

**Notation.** Let $\mathbf{X} = \{X_1, X_2, \ldots, X_n\}$ represent the $n$ random variables denoting the Key Performance Indicators (KPIs) for each node in a system. We use $\boldsymbol{x} = (x_1, \ldots, x_n)$ to denote their realizations. These nodes are interconnected via a topology defined by a graph $\mathcal{G} = (\mathbf{X}, \mathbf{E})$ that is assumed to be directed and acyclic. An edge $(X_i, X_j) \in \mathbf{E}$ indicates that changes to $X_i$ affect the values $X_j$ can take, but not vice versa. We denote the parents of node $i$ in the graph $\mathcal{G}$ as $\text{Pa}_{X_i}$, and the corresponding values assigned to these parents in $\boldsymbol{x}$ as $\text{Pa}_{x_i}$. We assume that each observed instance $x_i$ is generated from its causal parents and a node specific latent exogenous variable $\epsilon_i$ via a structural causal model (SCM) as outlined below.

**Structural Causal Models (SCM).** An SCM  (Pearl, 2019) is a four-tuple $\mathcal{S} = (\mathbf{X}, \boldsymbol{\epsilon}, \mathcal{F}, P_{\boldsymbol{\epsilon}})$, where $P_{\boldsymbol{\epsilon}}$ represents the distribution from which the exogenous variables are sampled. The observed variables $X_i \in \mathbf{X}$ are determined by a set of structural equations $\mathcal{F} = \{f_1, \ldots, f_n\}$, where $f_i : \text{Pa}_{X_i} \times \epsilon_i \mapsto X_i$. This implies that, conditioned on its immediate parents, no other variable can exert a causal influence on $X_i$—a principle known as *modularity* in causal inference. In summary, an SCM models a data-generating process where an observed instance $\boldsymbol{x}$ is produced by sampling the latent exogenous variables $\boldsymbol{\epsilon} \sim P_{\boldsymbol{\epsilon}}$ first, and then subsequently computing the node values by applying the structural equations in a topological order defined over the causal graph $\mathcal{G}$.

**Counterfactuals.** Given an observed instance $\boldsymbol{x} = (x_1, \ldots, x_n)$, we may want to generate a hypothetical instance $\boldsymbol{x}^{\text{CF}(j)}$ representing what $\boldsymbol{x}$ would have been if $X_j$ had taken the value $x_j'$ instead of $x_j$. This hypothetical instance, called a counterfactual, is computed using an SCM $\mathcal{S}$ in three steps:

1. *Abduction*: For each $i$, estimate $\hat{\epsilon}_i$ by inverting the function $f_i$ so that $x_i = f_i(\text{Pa}_{x_i}, \epsilon_i)$ holds in the SCM $\mathcal{S}$.
2. *Action*: Set $x_i^{\text{CF}(j)} = x_i$ for any $X_i$ that is not a descendant of $X_j$ in $\mathcal{G}$. For $X_j$, set $x_j^{\text{CF}(j)} = x_j'$.
3. *Prediction*: For each descendant $X_i$ of $X_j$, in topological order, set $x_i^{\text{CF}(j)} = f_i(\text{pa}_{x_i}^{\text{CF}(j)}, \hat{\epsilon}_i)$, using $\hat{\epsilon}_i$ obtained during the abduction step.

**Interventions.** An intervention $\boldsymbol{x}^{\text{int}(j)}$ represents the distributional effect of setting $X_j$ to $x_j'$ on its descendants in the causal graph $\mathcal{G}$. They are sampled as as follows:

1. Sample $\tilde{\epsilon}_i$ from its marginal distribution $P_{\epsilon_i}$.
2. Steps 2 and 3 are the same as for counterfactuals, except that interventions use the sampled value $\tilde{\epsilon}_i$ in place of the abducted $\hat{\epsilon}_i$ during prediction; i.e., $\boldsymbol{x}_i^{\text{int}(j)} = f_i(\text{pa}_{x_i}^{\text{int}(j)}, \tilde{\epsilon}_i)$.

Note that the abduction step in counterfactuals necessitates each $f_i$ to be invertible with respect to $\epsilon_i$, whereas interventions do not need this requirement. Further, while $\boldsymbol{x}^{\text{CF}(j)}$ is a point estimate, $\boldsymbol{x}^{\text{int}(j)}$ is a random variable due to the randomness in $\tilde{\epsilon}_i$.

## B   IDI ALGORITHM

We provide the pseudocode for IDI in Alg. 1.

## C   PROOFS

**Definition 8 ((Redko et al., 2019))** *For a loss function $\ell$ and hypothesis class $\mathcal{H}$, we define the discrepancy distance between two distributions $P, Q$ as follows:*

$$disc_\ell^{\mathcal{H}}(P, Q) = \sup_{h, h' \in \mathcal{H}^2} |\mathbb{E}_{x \sim P}[\ell(h(x), h'(x))] - \mathbb{E}_{x \sim Q}[\ell(h(x), h'(x))]|$$

**Lemma 9** *For bounded loss functions $\ell(\bullet, \bullet) \leq M$, where $M > 0$, we have:*

$$disc_\ell^{\mathcal{H}}(P, Q) \leq M tvd(P, Q)$$

See (Redko et al., 2019) (Proposition 3.1) for the proof.

---

**Algorithm 1** IDI Algorithm

---

**Require:** Data $D_{\text{trn}}$, DAG $\mathcal{G}$, anomalous instance $\boldsymbol{x}$, anomaly config $\mathcal{A}$
1: **Ensure** $\widehat{\mathcal{R}}$      ◁ predicted root causes
2: $\{\varphi_i, g_i\} \leftarrow \texttt{TrainAnomalyFns}(D_{\text{trn}}, \mathcal{G}, \mathcal{A})$    ◁ $\mathcal{A}$ is part of problem spec; default $g_i$ is Z-Score
3: $\mathcal{R}_{\text{cand}} \leftarrow \{i : \varphi_i(x_i) = 1, \text{ and } \varphi_p(x_p) = 0 \ \forall \, p \in \text{Pa}_{X_i}\}$    ◁ anomaly condition
4: $\widehat{\mathcal{S}} \leftarrow \text{FitSCM}(D_{\text{trn}}, \mathcal{G})$    ◁ Fits structural equation $\hat{f}_i$ for each $x_i$ on input $\text{Pa}_{x_i}$ using $L_2$ loss
5: **for** $\alpha \in 2^{\mathcal{R}_{\text{cand}}}$ **do**
6:     $\alpha_t \leftarrow \text{TopologicalSort}(\alpha, \mathcal{G})$    ◁ To apply the fixes in topological order
7:     Set $x_j^{\text{fix}} \sim P(X_j | \text{Pa}_{x_j}) \forall j \in \alpha_t$    ◁ Follow order $\alpha_t$
8:     $\widehat{\boldsymbol{x}}^{\text{int}(\alpha)} \leftarrow \text{Intervene}(\widehat{\mathcal{S}}, \boldsymbol{x}, \text{fix})$    ◁ Intervene on $\alpha$ using the SCM $\widehat{\mathcal{S}}$
9:     $\phi[\alpha] \leftarrow g_n(x_n) - \mathbb{E}[g_n(\widehat{x}_n^{\text{int}(\alpha)})]$    ◁ Shapley utility for the subset $\alpha$
10: **end for**
11: $\widehat{\mathcal{R}} \leftarrow \text{ShapleySort}(\phi, \text{ desc})$    ◁ Compute Shapley values and Sort $\widehat{\mathcal{R}}$ descending
12: **return** $\widehat{\mathcal{R}}$

---

**Theorem 5** *Suppose the true SCM $\mathcal{S}$ is an additive noise model defined over a chain graph $\mathcal{G} = X_1 \rightarrow \ldots \rightarrow X_n$ with structural equations of the form $f_i(x_{i-1}) + \epsilon_i$, where $\epsilon_i$ has bounded variance $\sigma^2$ and $f_i$ is $K$-Lipschitz. Let $\mathcal{H} = \{\mathcal{H}_i\}_{i=1}^n$ be the realizable hypothesis class for $\mathcal{S}$, with each $\mathcal{H}_i$ containing bounded functions that are $K$-lipschitz. Let $\widehat{\mathcal{S}}$ denote the SCM learned from training data $D_{\text{trn}}$. $\widehat{\mathcal{S}}$ encompasses the estimated functions $\{\hat{f}_i\}_{i=1}^n$. For any $j$, let $Q_X^{RC(j)}$ denote the distribution of samples with a unique root cause at $X_j$. Then, for $\boldsymbol{x}$ sampled from $Q_X^{RC(j)}$ with fix $x_j^{\text{fix}} \sim \hat{P}_X^{\text{trn}}(X_j | X_{j-1} = x_{j-1})$ applied to the root cause $X_j$, the error in estimated counterfactual at the target node $X_n$ admits the following bound:*

$$
\begin{aligned}
\mathbb{E}_{\boldsymbol{x} \sim Q_X^{RC(j)}}[|x_n^{CF(j)} - \widehat{x}_n^{CF(j)}|] \leq \sum_{i > j} K^{n-i} \bigg[ & 2^{n-i+1} \mathbb{E}_{x_{i-1} \sim P_{X_{i-1}}^{trn}} \left[ |f_i(x_{i-1}) - \hat{f}_i(x_{i-1})| \right] \\
& + M^{n-i+1} \cdot \left( tvd(P_{X_{i-1}}^{trn}, Q_{X_{i-1}}^{RC(j)}) + tvd(P_{X_{i-1}}^{trn}, P_{X_{i-1}}^{\widehat{trn}}) \right) \bigg]
\end{aligned}
\tag{10}
$$

**Proof** For the intervention $x_j^{\text{fix}}$ applied on $X_j$, let us denote the true counterfactual values obtained from the SCM $\mathcal{S}$ as $\boldsymbol{x}^{\text{CF}(j)}$, and the estimated counterfactual from the learned SCM $\widehat{\mathcal{S}}$ as $\widehat{\boldsymbol{x}}^{\text{CF}(j)}$. Then, at $j + 1$, we have:

$$
x_{j+1}^{\text{CF}(j)} = f_{j+1}(x_j^{\text{fix}}) + \epsilon_{j+1} \text{ where } \epsilon_{j+1} = x_{j+1} - f_{j+1}(x_j) \tag{11}
$$

$$
\widehat{x}_{j+1}^{\text{CF}(j)} = \hat{f}_{j+1}(x_j^{\text{fix}}) + \hat{\epsilon}_{j+1} \text{ where } \hat{\epsilon}_{j+1} = x_{j+1} - \hat{f}_{j+1}(x_j) \tag{12}
$$

Now, let us bound the error in the true CF $\boldsymbol{x}^{\text{CF}(j)}$ and the estimated CF $\widehat{\boldsymbol{x}}^{\text{CF}(j)}$ using $\widehat{\mathcal{S}}$ at index $j + 1$. Taking difference of Eqs. 11, 12

$$
|x_{j+1}^{\text{CF}(j)} - \widehat{x}_{j+1}^{\text{CF}(j)}| = |f_{j+1}(x_j^{\text{fix}}) - f_{j+1}(x_j) - \hat{f}_{j+1}(x_j^{\text{fix}}) + \hat{f}_{j+1}(x_j)| \tag{13}
$$

$$
\leq |f_{j+1}(x_j^{\text{fix}}) - \hat{f}_{j+1}(x_j^{\text{fix}})| + |f_{j+1}(x_j) - \hat{f}_{j+1}(x_j)| \tag{14}
$$

$$
\mathbb{E}_{\boldsymbol{x} \sim Q_X^{RC(j)}}[|x_{j+1}^{\text{CF}(j)} - \widehat{x}_{j+1}^{\text{CF}(j)}|] \leq \underbrace{\mathbb{E}_{\boldsymbol{x} \sim Q_X^{RC(j)}} \left[ \mathbb{E}_{x_j^{\text{fix}} \sim P_{X_j}^{\widehat{trn}}(X_j | x_{j-1})} \left[ |f_{j+1}(x_j^{\text{fix}}) - \hat{f}_{j+1}(x_j^{\text{fix}})| \right] \right]}_{\text{term 1}}
$$

$$
+ \underbrace{\mathbb{E}_{\boldsymbol{x} \sim Q_X^{RC(j)}}[|f_{j+1}(x_j) - \hat{f}_{j+1}(x_j)|]}_{\text{term 2}} \tag{15}
$$

Let us bound term 2 above.

Recall that $\widehat{S}$ is obtained by training from $D_{\text{trn}}$, $\hat{f}_{j+1}$ is learned using $N$ samples obtained i.i.d. from $P_{X_j}^{\text{trn}}$. Further, the loss is assessed only for $X_{j+1}$; we have $\mathbb{E}_{\boldsymbol{x} \sim Q_X^{\text{RC}(j)}}[|f_{j+1}(x_j) - \hat{f}_{j+1}(x_j)|] = \mathbb{E}_{x_j \sim Q_{X_j}^{\text{RC}(j)}}[|f_{j+1}(x_j) - \hat{f}_{j+1}(x_j)|]$.

Therefore, we have:

$$\mathbb{E}_{x_j \sim Q_{X_j}^{\text{RC}(j)}}[|f_{j+1}(x_j) - \hat{f}_{j+1}(x_j)|] \tag{16}$$

$$= \mathbb{E}_{x_j \sim Q_{X_j}^{\text{RC}(j)}}[|f_{j+1}(x_j) - \hat{f}_{j+1}(x_j)|] + \mathbb{E}_{x_j \sim P_{X_j}^{\text{trn}}}[|f_{j+1}(x_j) - \hat{f}_{j+1}(x_j)|]$$

$$\quad - \mathbb{E}_{x_j \sim P_{X_j}^{\text{trn}}}[|f_{j+1}(x_j) - \hat{f}_{j+1}(x_j)|]$$

$$\leq \mathbb{E}_{x_j \sim P_{X_j}^{\text{trn}}}[|f_{j+1}(x_j) - \hat{f}_{j+1}(x_j)|] + \int_{x_j} |f_{j+1}(x_j) - \hat{f}_{j+1}(x_j)| \cdot |P_{X_j}^{\text{trn}}(x_j) - Q_{X_j}^{\text{RC}(j)}(x_j)| dx_j \tag{17}$$

$$\leq \mathbb{E}_{x_j \sim P_{X_j}^{\text{trn}}}[|f_{j+1}(x_j) - \hat{f}_{j+1}(x_j)|] \tag{18}$$

$$\quad + \sup_{h,h' \in \mathcal{H}_{j+1}^2} \int_{x_j} |h(x_j) - h'(x_j)| \cdot |P_{X_j}^{\text{trn}}(x_j) - Q_{X_j}^{\text{RC}(j)}(x_j)| dx_j$$

$$= \mathbb{E}_{x_j \sim P_{X_j}^{\text{trn}}}[|f_{j+1}(x_j) - \hat{f}_{j+1}(x_j)|] + \text{disc}_{\ell_1}^{\mathcal{H}_{j+1}}(P_{X_j}^{\text{trn}}(x_j), Q_{X_j}^{\text{RC}(j)}(x_j)) \tag{19}$$

The last inequality is valid as long as $f_{j+1} \in \mathcal{H}_{j+1}$ which is true because $\mathcal{H}_{j+1}$ is realizable. Further since $\mathcal{H}_{j+1}$ encompasses bounded functions, we can bound the $L_1$ loss $\ell_1$ using a constant $M > 0$. Therefore,

$$\mathbb{E}_{\boldsymbol{x} \sim Q_{X_j}^{\text{RC}(j)}}[|f_{j+1}(x_j) - \hat{f}_{j+1}(x_j)|] \leq \underbrace{\mathbb{E}_{x_j \sim P_{X_j}^{\text{trn}}}[|f_{j+1}(x_j) - \hat{f}_{j+1}(x_j)|]}_{\text{term 1}} + \underbrace{M \text{tvd}(P_{X_j}^{\text{trn}}(x_j), Q_{X_j}^{\text{RC}(j)}(x_j))}_{\text{term 2}} \tag{20}$$

where term 1 represents a classical divergence between empirical risk and true risk, and can be bounded using classical VC, Rademacher complexity based generalization bounds. Term 2 is more interesting as it captures the divergence between the standard training distribution and the root cause distribution that governs when an anomaly occurs.

Combining Eqs. 15, 20 for the error at $X_{j+1}$, we get

$$\mathbb{E}_{\boldsymbol{x} \sim Q_X^{\text{RC}(j)}}[|x_{j+1}^{\text{CF}(j)} - \hat{x}_{j+1}^{\text{CF}(j)}|] \leq \underbrace{\mathbb{E}_{\boldsymbol{x} \sim Q_X^{\text{RC}(j)}}\left[\mathbb{E}_{x_j^{\text{fix}} \sim P_{X_j}^{\hat{\text{trn}}}(X_j|x_{j-1})}\left[|f_{j+1}(x_j^{\text{fix}}) - \hat{f}_{j+1}(x_j^{\text{fix}})|\right]\right]}_{\text{term 1}} \tag{21}$$

$$+ \underbrace{\mathbb{E}_{x_j \sim P_{X_j}^{\text{trn}}}[|f_{j+1}(x_j) - \hat{f}_{j+1}(x_j)|] + M \cdot \text{tvd}(P_{X_j}^{\text{trn}}, Q_{X_j}^{\text{RC}(j)})}_{\text{term 2}}$$

Now, we reduce the first term above. For a fix $x_j^{\text{fix}} \sim P_{X_j}^{\hat{\text{trn}}}(X_j|x_{j-1})$, the following holds:

$$\mathbb{E}_{\boldsymbol{x} \sim Q_X^{\text{RC}(j)}}\left[\mathbb{E}_{x_j^{\text{fix}} \sim P_{X_j}^{\hat{\text{trn}}}(X_j|x_{j-1})}\left[|f_{j+1}(x_j^{\text{fix}}) - \hat{f}_{j+1}(x_j^{\text{fix}})|\right]\right] \tag{22}$$

$$= \mathbb{E}_{\boldsymbol{x} \sim Q_X^{\text{RC}(j)}}\left[\mathbb{E}_{x_j^{\text{fix}} \sim P_{X_j}^{\hat{\text{trn}}}(X_j|x_{j-1})}\left[|f_{j+1}(x_j^{\text{fix}}) - \hat{f}_{j+1}(x_j^{\text{fix}})|\right]\right]$$

$$+ \mathbb{E}_{\boldsymbol{x} \sim Q_X^{\text{RC}(j)}}\left[\mathbb{E}_{x_j^{\text{fix}} \sim P_{X_j}^{\text{trn}}(X_j|x_{j-1})}\left[|f_{j+1}(x_j^{\text{fix}}) - \hat{f}_{j+1}(x_j^{\text{fix}})|\right]\right]$$

$$- \mathbb{E}_{\boldsymbol{x} \sim Q_X^{\text{RC}(j)}}\left[\mathbb{E}_{x_j^{\text{fix}} \sim P_{X_j}^{\text{trn}}(X_j|x_{j-1})}\left[|f_{j+1}(x_j^{\text{fix}}) - \hat{f}_{j+1}(x_j^{\text{fix}})|\right]\right]$$

$$\leq \mathbb{E}_{x_j \sim P_{X_j}^{\text{trn}}}\left[|f_{j+1}(x_j) - \hat{f}_{j+1}(x_j)|\right] + M \cdot \text{tvd}(P_{X_j}^{\text{trn}}, P_{X_j}^{\hat{\text{trn}}}) \tag{23}$$

This follows from the definition of $Q^{\text{RC}(j)}$ 1. First observe that for the indices till $\{1, \ldots, j-1\}$, both root cause distribution $Q_X^{\text{RC}(j)}$ and the training distribution $P_X^{\text{trn}}$ agree; i.e., $Q_X^{\text{RC}(j)}(X_1, \ldots, X_{j-1}) = P_X^{\text{trn}}(X_1, \ldots, X_{j-1})$. The only index that changes is $j$ at which the root cause occurs. But since, while applying a fix, we sample it from $P_X^{\text{trn}}(X_j | x_{j-1})$, the marginal distribution of the fix simply reduces to $P_{X_j}^{\text{trn}}$.

Finally, combining the inequalities in 21, 23, we get:

$$
\begin{aligned}
\mathbb{E}_{x_j \sim Q_X^{\text{RC}(j)}}[|x_{j+1}^{\text{CF}(j)} - \widehat{x}_{j+1}^{\text{CF}(j)}|] \leq{} & 2\mathbb{E}_{x_j \sim P_{X_j}^{\text{trn}}}[|f_{j+1}(x_j) - \hat{f}_{j+1}(x_j)|] \\
& + M \cdot \left[ \text{tvd}(P_{X_j}^{\text{trn}}, Q_{X_j}^{\text{RC}(j)}) + \text{tvd}(P_{X_j}^{\text{trn}}, P_{X_j}^{\widehat{\text{trn}}}) \right]
\end{aligned}
\tag{24}
$$

Now, let us carry forward these arguments to $j + 2$.

$$
x_{j+2}^{\text{CF}(j)} = f_{j+2}(x_{j+1}^{\text{CF}(j)}) + \epsilon_{j+2} \text{ where } \epsilon_{j+2} = x_{j+2} - f_{j+2}(x_{j+1})
\tag{25}
$$

$$
\widehat{x}_{j+2}^{\text{CF}(j)} = \hat{f}_{j+2}(\widehat{x}_{j+1}^{\text{CF}(j)}) + \hat{\epsilon}_{j+2} \text{ where } \hat{\epsilon}_{j+2} = x_{j+2} - \hat{f}_{j+2}(x_{j+1})
\tag{26}
$$

$$
\tag{27}
$$

Now for the estimated counterfactual at $X_{j+2}$, we have the error

$$
|x_{j+2}^{\text{CF}(j)} - \widehat{x}_{j+2}^{\text{CF}(j)}| = |f_{j+2}(x_{j+1}^{\text{CF}(j)}) - f_{j+2}(x_{j+1}) - \hat{f}_{j+2}(\widehat{x}_{j+1}^{\text{CF}(j)}) + \hat{f}_{j+2}(x_{j+1})|
\tag{28}
$$

$$
\leq \underbrace{|f_{j+2}(x_{j+1}^{\text{CF}(j)}) - \hat{f}_{j+2}(\widehat{x}_{j+1}^{\text{CF}(j)})|}_{\text{term 1}} + \underbrace{|f_{j+2}(x_{j+1}) - \hat{f}_{j+2}(x_{j+1})|}_{\text{term 2}}
\tag{29}
$$

The term 2 above admits the same proof technique as we derived for $X_{j+1}$.

$$
\begin{aligned}
\mathbb{E}_{\boldsymbol{x} \sim Q_X^{\text{RC}(j)}}[|f_{j+2}(x_{j+1}) - \hat{f}_{j+2}(x_{j+1})|] \leq{} & \mathbb{E}_{x_{j+1} \sim P_{X_{j+1}}^{\text{trn}}}[|f_{j+2}(x_{j+1}) - \hat{f}_{j+2}(x_{j+1})|] \\
& + M \cdot \text{tvd}(P_{X_{j+1}}^{\text{trn}}, Q_{X_{j+1}}^{\text{RC}(j)})
\end{aligned}
\tag{30}
$$

Now, let us analyze the first term.

Since $\hat{f}_{j+2}$ is $K$-Lipschitz, we have $|\hat{f}_{j+2}(\widehat{x}_{j+1}^{\text{CF}(j)}) - \hat{f}_{j+2}(x_{j+1}^{\text{CF}(j)})| \leq K|\widehat{x}_{j+1}^{\text{CF}(j)} - x_{j+1}^{\text{CF}(j)}|$. Therefore,

$$
\mathbb{E}_{\boldsymbol{x} \sim Q_X^{\text{RC}(j)}}[|f_{j+2}(x_{j+1}^{\text{CF}(j)}) - \hat{f}_{j+2}(\widehat{x}_{j+1}^{\text{CF}(j)})|]
\tag{31}
$$

$$
= \mathbb{E}_{\boldsymbol{x} \sim Q_X^{\text{RC}(j)}}[|f_{j+2}(x_{j+1}^{\text{CF}(j)}) - \hat{f}_{j+2}(x_{j+1}^{\text{CF}(j)}) + \hat{f}_{j+2}(x_{j+1}^{\text{CF}(j)}) - \hat{f}_{j+2}(\widehat{x}_{j+1}^{\text{CF}(j)})|]
$$

$$
\leq \mathbb{E}_{\boldsymbol{x} \sim Q_X^{\text{RC}(j)}}[|f_{j+2}(x_{j+1}^{\text{CF}(j)}) - \hat{f}_{j+2}(x_{j+1}^{\text{CF}(j)})|] + \mathbb{E}_{\boldsymbol{x} \sim Q_X^{\text{RC}(j)}}[|\hat{f}_{j+2}(x_{j+1}^{\text{CF}(j)}) - \hat{f}_{j+2}(\widehat{x}_{j+1}^{\text{CF}(j)})|]
\tag{32}
$$

$$
\leq \mathbb{E}_{\boldsymbol{x} \sim Q_X^{\text{RC}(j)}}[|f_{j+2}(x_{j+1}^{\text{CF}(j)}) - \hat{f}_{j+2}(x_{j+1}^{\text{CF}(j)})|] + K\mathbb{E}_{\boldsymbol{x} \sim Q_X^{\text{RC}(j)}}[|x_{j+1}^{\text{CF}(j)} - \widehat{x}_{j+1}^{\text{CF}(j)}|]
\tag{33}
$$

Using the same arguments as used in Eq. 23, it can be shown that:

$$
\begin{aligned}
\mathbb{E}_{\boldsymbol{x} \sim Q_X^{\text{RC}(j)}}[|f_{j+2}(x_{j+1}^{\text{CF}(j)}) - \hat{f}_{j+2}(x_{j+1}^{\text{CF}(j)})|] \leq{} & \mathbb{E}_{x_{j+1} \sim P_{X_{j+1}}^{\text{trn}}}[|f_{j+2}(x_{j+1}) - \hat{f}_{j+2}(x_{j+1})|] \\
& + M \cdot \text{tvd}(P_{X_{j+1}}^{\text{trn}}, P_{X_{j+1}}^{\widehat{\text{trn}}})
\end{aligned}
\tag{34}
$$

Finally, we have:

$$\mathbb{E}_{\boldsymbol{x}\sim Q_X^{\text{RC}(j)}}[|x_{j+2}^{\text{CF}(j)} - \widehat{x}_{j+2}^{\text{CF}(j)}|] \leq 2\mathbb{E}_{x_{j+1}\sim P_{X_{j+1}}^{\text{trn}}}[|f_{j+2}(x_{j+1}) - \hat{f}_{j+2}(x_{j+1})|] \quad (35)$$
$$+ K\mathbb{E}_{\boldsymbol{x}\sim Q_X^{\text{RC}(j)}}[|x_{j+1}^{\text{CF}(j)} - \widehat{x}_{j+1}^{\text{CF}(j)}|]$$
$$+ M \cdot \left[\text{tvd}(P_{X_{j+1}}^{\text{trn}}, Q_{X_{j+1}}^{\text{RC}(j)}) + \text{tvd}(P_{X_{j+1}}^{\text{trn}}, P_{X_{j+1}}^{\widehat{\text{trn}}})\right]$$

Now, we can extend this result to assess the error at $X_n$ as follows:

$$\mathbb{E}_{\boldsymbol{x}\sim Q_X^{\text{RC}(j)}}[|x_n^{\text{CF}(j)} - \widehat{x}_n^{\text{CF}(j)}|] \leq \sum_{i>j} K^{n-i}\left[2^{n-i+1}\mathbb{E}_{x_{i-1}\sim P_{X_{i-1}}^{\text{trn}}}\left[|f_i(x_{i-1}) - \hat{f}_i(x_{i-1})|\right]\right.$$
$$\left. + M^{n-i+1} \cdot \left(\text{tvd}(P_{X_{i-1}}^{\text{trn}}, Q_{X_{i-1}}^{\text{RC}(j)}) + \text{tvd}(P_{X_{i-1}}^{\text{trn}}, P_{X_{i-1}}^{\widehat{\text{trn}}})\right)\right] \quad (36)$$

The above inequality holds because from definition of $Q_X^{\text{RC}(j)}$ 1, we have $\mathbb{E}_{\boldsymbol{x}\sim Q_X^{\text{RC}(j)}}[|x_i^{\text{CF}(j)} - \widehat{x}_i^{\text{CF}(j)}|] = 0$ for any $i < j$. Further at $X_j$, because of performing an intervention we have $x_j^{\text{CF}(j)} = \widehat{x}_j^{\text{CF}(j)} = x_j^{\text{fix}}$.

∎

**Remark:** Several prior works (Chen et al., 2014; Lin et al., 2018; Liu et al., 2021) defined root cause as a node that is anomalous and is connected to target node $X_n$ through a chain of anomalous nodes. i.e., they expect $\varphi_i(x_i) = 1$ for all $i \geq j$, and in such a case, we see that $\text{tvd}\left(P_{X_i}^{\text{trn}}, Q_{X_i}^{\text{RC}(j)}\right)$ is large for any $i > j$. Nonetheless, the leading term in the above geometric progression has $\text{tvd}\left(P_{X_j}^{\text{trn}}, Q_{X_k}^{\text{RC}(j)}\right)$ for which we know $\varphi_j(x_j) = 1$ by root cause definition, and therefore, in practice $\widehat{\boldsymbol{x}}^{\text{CF}(j)}$ is a poor estimate for $\boldsymbol{x}^{\text{CF}(j)}$. All the other terms can be bounded using classical generalization bounds, and they go down with the size of training data $|D_{\text{trn}}|$.

**Theorem 6** *Under the same conditions laid out in Theorem 5, the error between the true counterfactual $\boldsymbol{x}^{CF(j)}$ and the estimated intervention $\widehat{\boldsymbol{x}}^{int(j)}$ admits the following bound:*

$$\mathbb{E}_{\boldsymbol{x}\sim Q_X^{RC(j)}}\left[|x_n^{CF(j)} - \widehat{x}_n^{int(j)}|\right] \leq \sum_{i>j} K^{n-i}\left[\mathbb{E}_{x_{i-1}\sim P_{X_{i-1}}^{trn}}\left[|f_i(x_{i-1}^{CF(j)}) - \hat{f}_i(x_{i-1}^{CF(j)})|\right]\right.$$
$$+ M^{n-i+1}\, \text{tvd}\left(P_{X_{i-1}}^{trn}, \hat{P}^{trn}_{X_{i-1}}\right) + \text{std}(\epsilon_i)$$
$$\left. + \text{std}(\tilde{\epsilon}_i) + \left|\mathbb{E}[\epsilon_i] - \mathbb{E}[\tilde{\epsilon}_i]\right|\right]$$

$$(37)$$

*where* $bias(\hat{f}_i) = \mathbb{E}_{x_i\sim P_{X_i}^{trn}}[f_i(x_i) - \hat{f}_i(x_i)]$. *For an unbiased learner, bias is zero.*

**Proof** For the intervention $x_j^{\text{fix}}$ applied on $X_j$, let us denote the true counterfactual values obtained from the SCM $\mathcal{S}$ as $\boldsymbol{x}^{\text{CF}(j)}$, and the estimated intervention from the learned SCM $\widehat{\mathcal{S}}$ as $\widehat{\boldsymbol{x}}^{\text{int}(j)}$. Then, at $j+1$, we have:

$$x_{j+1}^{\text{CF}(j)} = f_{j+1}(x_j^{\text{fix}}) + \epsilon_{j+1} \text{ where } \epsilon_{j+1} = x_{j+1} - f_{j+1}(x_j) \quad (38)$$
$$\widehat{x}_{j+1}^{\text{int}(j)} = \hat{f}_{j+1}(x_j^{\text{fix}}) + \tilde{\epsilon}_{j+1} \text{ where } \tilde{\epsilon}_{j+1} \sim P_{\epsilon_{j+1}}^{\widehat{\text{trn}}} \quad (39)$$

where, $P_{\epsilon_{j+1}}^{\widehat{\text{trn}}}$ represents the empirical distribution of the marginal $P_{\epsilon_{j+1}}^{\text{trn}}$, obtained from a validation dataset $D_V \subset D_{\text{trn}}$. For additive noise models, this is simply the empirical distribution of the error residuals, $x_{j+1} - \hat{f}_{j+1}(x_j)$ for $\boldsymbol{x} \in D_V$.

Now, let us bound the error in the true CF $x^{\text{CF}(j)}$ and the estimated intervention $\widehat{x}^{\text{int}(j)}$ using $\widehat{S}$ at index $j+1$. Taking difference of above Eqs.

$$\mathbb{E}_{\boldsymbol{x} \sim Q_X^{\text{RC}(j)}}[|x_{j+1}^{\text{CF}(j)} - \widehat{x}_{j+1}^{\text{int}(j)}|] = \underbrace{\mathbb{E}_{\boldsymbol{x} \sim Q_X^{\text{RC}(j)}} \left[ \mathbb{E}_{x_j^{\text{fix}} \sim P_{X_j}^{\widehat{\text{trn}}}(X_j | x_{j-1})} \left[ |f_{j+1}(x_j^{\text{fix}}) - \hat{f}_{j+1}(x_j^{\text{fix}})| \right] \right]}_{\text{term 1}} \quad (40)$$
$$+ \underbrace{\mathbb{E}_{\boldsymbol{x} \sim Q_X^{\text{RC}(j)}} \mathbb{E}_{\tilde{\epsilon}_{j+1} \sim \hat{\mathcal{P}}_{\epsilon_{j+1}}} [|\epsilon_{j+1} - \tilde{\epsilon}_{j+1}|]}_{\text{term 2}}$$

We use Eq. 23 to reduce term 1 to $\mathbb{E}_{x_j \sim P_{X_j}^{\text{trn}}} \left[ |f_{j+1}(x_j) - \hat{f}_{j+1}(x_j)| \right] + M \cdot \text{tvd}(P_{X_j}^{\text{trn}}, P_{X_j}^{\widehat{\text{trn}}})$.

Now, let us analyze term 2. Since $X_j$ is the unique root cause, we have $Q_{\epsilon_{j+1}}^{\text{RC}(j)} = P_{\epsilon_{j+1}}^{\text{trn}}$. Therefore,

$$\mathbb{E}_{\epsilon_{j+1} \sim Q_{\epsilon_{j+1}}^{\text{RC}(j)}} \mathbb{E}_{\tilde{\epsilon}_{j+1} \sim \hat{\mathcal{P}}_{\epsilon_{j+1}}} [|\epsilon_{j+1} - \mathbb{E}[\tilde{\epsilon}_{j+1}]|] = \mathbb{E}_{\epsilon_{j+1} \sim P_{\epsilon_{j+1}}^{\text{trn}}} \mathbb{E}_{\tilde{\epsilon}_{j+1} \sim \hat{\mathcal{P}}_{\epsilon_{j+1}}} [|\epsilon_{j+1} - \mathbb{E}[\tilde{\epsilon}_{j+1}]|] \quad (41)$$

Now,

$$|\epsilon_{j+1} - \tilde{\epsilon}_{j+1}| = |\epsilon_{j+1} - \mathbb{E}[\epsilon_{j+1}] + \mathbb{E}[\epsilon_{j+1}] - \mathbb{E}[\tilde{\epsilon}_{j+1}] + \mathbb{E}[\tilde{\epsilon}_{j+1}] - \tilde{\epsilon}_{j+1}| \quad (42)$$
$$\leq |\epsilon_{j+1} - \mathbb{E}[\epsilon_{j+1}]| + |\mathbb{E}[\epsilon_{j+1}] - \mathbb{E}[\tilde{\epsilon}_{j+1}]| + |\mathbb{E}[\tilde{\epsilon}_{j+1}] - \tilde{\epsilon}_{j+1}| \quad (43)$$

using $|a + b| \leq |a| + |b|$. Note that $|\mathbb{E}[\epsilon_{j+1}] - \mathbb{E}[\tilde{\epsilon}_{j+1}]|$ is a constant.

Taking expectation on both sides, we get:

$$\mathbb{E}_{\epsilon_{j+1} \sim P_{\epsilon_{j+1}}^{\text{trn}}} \mathbb{E}_{\tilde{\epsilon}_{j+1} \sim \hat{\mathcal{P}}_{\epsilon_{j+1}}} [|\epsilon_{j+1} - \mathbb{E}[\tilde{\epsilon}_{j+1}]|] \quad (44)$$
$$\leq \mathbb{E}_{\epsilon_{j+1} \sim P_{\epsilon_{j+1}}^{\text{trn}}} [|\epsilon_{j+1} - \mathbb{E}[\epsilon_{j+1}]|] + \mathbb{E}_{\tilde{\epsilon}_{j+1} \sim \hat{\mathcal{P}}_{\epsilon_{j+1}}} [|\mathbb{E}[\tilde{\epsilon}_{j+1}] - \tilde{\epsilon}_{j+1}| + |\mathbb{E}[\epsilon_{j+1}] - \mathbb{E}[\tilde{\epsilon}_{j+1}]|$$
$$\leq \sqrt{\mathbb{E}_{\epsilon_{j+1} \sim P_{\epsilon_{j+1}}^{\text{trn}}} [(\epsilon_{j+1} - \mathbb{E}[\epsilon_{j+1}])^2]} + \sqrt{\mathbb{E}_{\tilde{\epsilon}_{j+1} \sim \hat{\mathcal{P}}_{\epsilon_{j+1}}} [(\mathbb{E}[\tilde{\epsilon}_{j+1}] - \tilde{\epsilon}_{j+1})^2]} + |\mathbb{E}[\epsilon_{j+1}] - \mathbb{E}[\tilde{\epsilon}_{j+1}]|$$
$$(45)$$
$$= \text{std}(\epsilon_{j+1}) + \text{std}(\tilde{\epsilon}_{j+1}) + |\mathbb{E}[\epsilon_{j+1}] - \mathbb{E}[\tilde{\epsilon}_{j+1}]| \quad (46)$$

where $\text{std}(\epsilon_{j+1})$ is the standard deviation of the latent noise $\epsilon_{j+1}$.

**Remark**: It is common in causal literature to assume that $\epsilon_i$ are zero-mean in addition to having bounded variance. When zero mean assumption holds, we can use $\tilde{\epsilon}_{j+1} = 0$, and in that case, we will have $\mathbb{E}_{\epsilon \sim Q_{\epsilon_{j+1}}^{\text{RC}(j)}} \mathbb{E}_{\tilde{\epsilon}_{j+1} \sim \hat{\mathcal{P}}_{\epsilon_{j+1}}} [|\epsilon_{j+1} - \mathbb{E}[\tilde{\epsilon}_{j+1}]|] \leq \sigma(\epsilon_{j+1})$.

Finally, for the estimated intervention at $X_j$, we have:

$$\boxed{\begin{aligned} \mathbb{E}_{\boldsymbol{x} \sim Q_X^{\text{RC}(j)}}[|x_{j+1}^{\text{CF}(j)} - \widehat{x}_{j+1}^{\text{int}(j)}|] &\leq \mathbb{E}_{x_j \sim P_{X_j}^{\text{trn}}} \left[ |f_{j+1}(x_j) - \hat{f}_{j+1}(x_j)| \right] \\ &+ M \cdot \text{tvd}(P_{X_j}^{\text{trn}}, P_{X_j}^{\widehat{\text{trn}}}) + \text{std}(\epsilon_{j+1}) + \text{std}(\tilde{\epsilon}_{j+1}) + |\mathbb{E}[\epsilon_{j+1}] - \mathbb{E}[\tilde{\epsilon}_{j+1}]| \end{aligned}} \quad (47)$$

For the estimated intervention at $X_{j+2}$, we have:

$$x_{j+2}^{\text{CF}(j)} = f_{j+2}(x_{j+1}^{\text{CF}(j)}) + \epsilon_{j+2} \text{ where } \epsilon_{j+2} = x_{j+2} - f_{j+2}(x_{j+1}) \quad (48)$$
$$\widehat{x}_{j+2}^{\text{int}(j)} = \hat{f}_{j+2}(\widehat{x}_{j+1}^{\text{int}(j)}) + \mathbb{E}[\tilde{\epsilon}_{j+2}] \text{ where } \tilde{\epsilon}_{j+2} \sim P_{\epsilon_{j+2}}^{\widehat{\text{trn}}} \quad (49)$$

Therefore, the error at $j+2$ is:

$$\mathbb{E}_{\boldsymbol{x} \sim Q_X^{\text{RC}(j)}}[|x_{j+2}^{\text{CF}(j)} - \widehat{x}_{j+2}^{\text{int}(j)}|] = \underbrace{\mathbb{E}_{\boldsymbol{x} \sim Q_X^{\text{RC}(j)}} \left[ |f_{j+2}(x_{j+1}^{\text{CF}(j)}) - \hat{f}_{j+2}(\widehat{x}_{j+1}^{\text{int}(j)})| \right]}_{\text{term 1}} \quad (50)$$
$$+ \underbrace{\mathbb{E}_{\boldsymbol{x} \sim Q_X^{\text{RC}(j)}}[|\epsilon_{j+2} - \mathbb{E}[\tilde{\epsilon}_{j+2}]|]}_{\text{term 2}}$$

We know that we can bound $\mathbb{E}_{\boldsymbol{x} \sim Q_X^{\mathrm{RC}(j)}}[|\epsilon_{j+2} - \mathbb{E}[\tilde{\epsilon}_{j+2}]|] \leq \mathrm{std}(\epsilon_{j+2}) + \mathrm{std}(\tilde{\epsilon}_{j+2}) + |\mathbb{E}[\epsilon_{j+2}] - \mathbb{E}[\tilde{\epsilon}_{j+2}]|$.

To bound the first term, we use Lipschitz property.

$$\mathbb{E}_{\boldsymbol{x} \sim Q_X^{\mathrm{RC}(j)}}[|f_{j+2}(x_{j+1}^{\mathrm{CF}(j)}) - \hat{f}_{j+2}(x_{j+1}^{\mathrm{CF}(j)})|] \leq \mathbb{E}_{\boldsymbol{x} \sim Q_X^{\mathrm{RC}(j)}}[|f_{j+2}(x_{j+1}^{\mathrm{CF}(j)}) - \hat{f}_{j+2}(x_{j+1}^{\mathrm{CF}(j)})|] \quad (51)$$
$$+ K \mathbb{E}_{\boldsymbol{x} \sim Q_X^{\mathrm{RC}(j)}}[|x_{j+1}^{\mathrm{CF}(j)} - \widehat{x}_{j+1}^{\mathrm{int}(j)}|]$$
$$\leq \mathbb{E}_{x_{j+1} \sim P_{X_{j+1}}^{\mathrm{trn}}}\left[|f_{j+2}(x_{j+1}) - \hat{f}_{j+2}(x_{j+1})|\right] \quad (52)$$
$$+ M \cdot \mathrm{tvd}(P_{X_{j+1}}^{\mathrm{trn}}, P_{X_{j+1}}^{\widehat{\mathrm{trn}}})$$

Finally, we have:

$$\mathbb{E}_{\boldsymbol{x} \sim Q_X^{\mathrm{RC}(j)}}[|x_{j+2}^{\mathrm{CF}(j)} - \widehat{x}_{j+2}^{\mathrm{int}(j)}|] \leq \mathbb{E}_{\boldsymbol{x} \sim Q_X^{\mathrm{RC}(j)}}[|f_{j+2}(x_{j+1}^{\mathrm{CF}(j)}) - \hat{f}_{j+2}(x_{j+1}^{\mathrm{CF}(j)})|] \quad (53)$$
$$+ K \mathbb{E}_{\boldsymbol{x} \sim Q_X^{\mathrm{RC}(j)}}[|x_{j+1}^{\mathrm{CF}(j)} - \widehat{x}_{j+1}^{\mathrm{int}(j)}|]$$
$$+ \mathrm{std}(\epsilon_{j+2}) + \mathrm{std}(\tilde{\epsilon}_{j+2}) + |\mathbb{E}[\epsilon_{j+2}] - \mathbb{E}[\tilde{\epsilon}_{j+2}]|$$

We can extend this result ts assess the error at $X_n$ as follows:

$$\mathbb{E}_{\boldsymbol{x} \sim Q_X^{\mathrm{RC}(j)}}\left[|x_n^{\mathrm{CF}(j)} - \widehat{x}_n^{\mathrm{int}(j)}|\right] \leq \sum_{i>j} K^{n-i}\left[\mathbb{E}_{x_{i-1} \sim P_{X_{i-1}}^{\mathrm{trn}}}\left[|f_i(x_{i-1}^{\mathrm{CF}(j)}) - \hat{f}_i(x_{i-1}^{\mathrm{CF}(j)})|\right]\right.$$
$$+ M^{n-i+1} \mathrm{tvd}\left(P_{X_{i-1}}^{\mathrm{trn}}, \hat{P}^{\mathrm{trn}}{}_{X_{i-1}}\right) + \mathrm{std}(\epsilon_i)$$
$$\left. + \mathrm{std}(\tilde{\epsilon}_i) + |\mathbb{E}[\epsilon_i] - \mathbb{E}[\tilde{\epsilon}_i|\right]$$
$$(54)$$

∎

**Corollary 7** *Suppose in Theorem 6, the exogenous variables $\epsilon_i$ are zero-mean in addition to having bounded variance for any $i \in [n]$, then the error between interventions and counterfactuals admits the following bound:*

$$\mathbb{E}_{\boldsymbol{x} \sim Q_X^{RC(j)}}[|x_n^{CF(j)} - \widehat{x}_n^{int(j)}|] \leq \sum_{i>j} K^{n-i}\left[\mathbb{E}_{x_{i-1} \sim P_{X_{i-1}}^{trn}}\left[|f_i(x_{i-1}^{CF(j)}) - \hat{f}_i(x_{i-1}^{CF(j)})|\right]\right.$$
$$\left. + M^{n-i+1} \mathrm{tvd}\left(P_{X_{i-1}}^{trn}, \hat{P}^{trn}{}_{X_{i-1}}\right) + \mathrm{std}(\epsilon_i)\right] \quad (55)$$

**Proof** If we know that the latent exogenous variables are zero-mean, we can set $P_{\epsilon_{j+1}}^{\widehat{\mathrm{trn}}}$ in Eq. 39 to the dirac-delta distribution $\delta(\tilde{\epsilon}_{j+1} = 0)$.

Then we can see from the remark below Eq. 46 that the proof simply follows. ∎

## D  TOY EXPERIMENTS

Section 6.1 outlined the experiments on toy datasets. Here, we provide a detailed description of the dataset generation process and conduct additional experiments to further highlight the differences between interventional and counterfactual estimates for RCD.

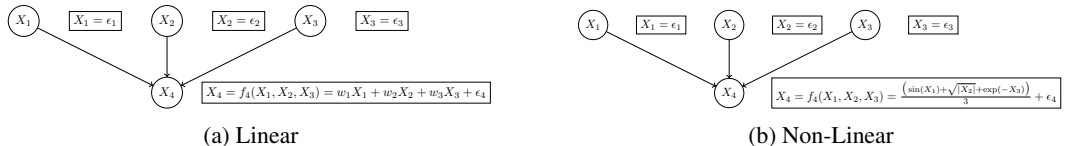

(a) Linear          (b) Non-Linear

Figure 6: This figure illustrates a toy causal graph with four variables: root nodes $X_1$, $X_2$, and $X_3$, and a child node $X_4$, which is connected to all three root nodes. Panel (a) depicts a linear SCM where the root nodes are direct copies of the exogenous noise terms $\epsilon$, while $X_4$ is a linear function of the root nodes. Panel (b) presents a non-linear version, where $X_4$ is the average of three non-linear functions applied to the root nodes. In both panels, the SCM follows an additive noise model.

Figure 7: Four variable toy SCMs.

|  | $X_1$ | $X_2$ | $X_3$ | $X_4$ |
|---|---|---|---|---|
| Training $\boldsymbol{x}^{\text{trn}}$ | $\mathcal{N}(0,1)$ | $\mathcal{N}(0,1)$ | $\mathcal{N}(0,1)$ | $f_4(\boldsymbol{x}^{\text{trn}}_{1:3}) + \mathcal{N}(0,\sigma^2)$ |
| Validation $\boldsymbol{x}^{\text{val}}$ | $\mathcal{N}(0,1)$ | $\mathcal{N}(0,1)$ | $\mathcal{N}(0,1)$ | $f_4(\boldsymbol{x}^{\text{val}}_{1:3}) + \mathcal{N}(0,\sigma^2)$ |
| Test $\boldsymbol{x}^{\text{RC}}$ | $\mathcal{N}(0,1)$ | $U[3,10]$ | $\mathcal{N}(0,1)$ | $f_4(\boldsymbol{x}^{\text{tst}}_{1:3}) + \mathcal{N}(0,\sigma^2)$ |
| Counterfactual $\widehat{\boldsymbol{x}}^{\text{CF}(2)}$ | $x^{\text{RC}}_1$ | $\mathcal{N}(0,1)$ | $x^{\text{RC}}_3$ | $f_4(\widehat{\boldsymbol{x}}^{\text{CF}(2)}_{1:3}) + \hat{\epsilon}_4$ |
| Intervention $\widehat{\boldsymbol{x}}^{\text{int}(2)}$ | $x^{\text{RC}}_1$ | $\mathcal{N}(0,1)$ | $x^{\text{RC}}_3$ | $f_4(\widehat{\boldsymbol{x}}^{\text{int}(2)}_{1:3}) + \tilde{\epsilon}_4$ |

Table 4: This table outlines the process for generating each sample in our toy simulation study. The node $X_4$ draws its $\epsilon_4$ from a normal distribution $\mathcal{N}(0,\sigma^2)$, with $\sigma^2$ varied between 0.5 and 5 in our experiments. For the test sample shown here, the root cause lies at $X_2$, and the corresponding $\epsilon_2$ is sampled from uniform $U[3,10]$. The fix $x^{\text{fix}}_2$ is sampled from $\mathcal{N}(0,1)$. For counterfactuals, $\hat{\epsilon}_4$ represents the abducted $\epsilon$, while for interventions, $\tilde{\epsilon}_4$ corresponds to a randomly sampled error residual obtained from the validation data.

We illustrate the four-variable toy SCM for both linear and non-linear cases in Figure 7. Table 4 presents the equations used to generate the toy dataset samples.

We now provide a detailed analysis of the key observations from the results shown in Figure 5 in our main paper.

- **Low Variance:** (a) A linear model, being convex, generalizes well across both training and test domains. As a result, the abducted values of $\hat{\epsilon}_4$ closely approximate the true values. This trend is evident in Fig. 5a, where the interventional error plateaus around the standard deviation 0.5, while the counterfactual (CF) error decreases with more training. (b) In contrast, for the non-linear model, interventional estimates significantly outperform CF estimates (note the $\log$ scale on the Y-axis). Nonlinear models are non-convex, leading to local minima or overfitting during training. Consequently, while validation error decreases rapidly, the test (RC) error remains high as training increases. This results in poor CF accuracy when $\hat{f}_4$ is evaluated on out-of-distribution (OOD) inputs.
- **Medium Variance:** Similar trends are observed in this setting. Figs. 5a and 5b together show that CF error correlates strongly with OOD test (RC) error, while interventional estimates remain bounded by the standard deviation. These results confirm the tightness of our theoretical bounds in practice.
- **High Variance:** In the linear dataset, the trends remain consistent. However, in the nonlinear dataset, interventional estimates show more pronounced advantages over CFs, especially in low-data regimes. This is because, in such scenarios, high variance $\epsilon_4$ destabilizes $\hat{f}_4$'s training, leading to high variance in its parameters. Only with sufficient training samples (e.g., 1000) does the validation error approach zero, causing the CF error to be comparable with the interventional error.
- **Very High Variance:** In this regime, we observe the first instance where CF error outperforms interventional error for the nonlinear model. One interpretation of the extreme variance in $\epsilon_4$ is that the features $X_1, X_2, X_3$ collected in the training data are insufficiently rich to explain the variance of $X_4$. This means that certain additional features must be collected, and thereby reduce the variance $\sigma^2$. Moreover, CFs perform better in this scenario only because their assumption of

additive noise holds. We will demonstrate in the subsequent experiment that CFs perform very poorly when this assumption is violated.
- **Summary:** While interventions may appear less favorable for linear models, we observed them to be adequate for in the root cause diagnosis problem. This is because, even if the estimated $\widehat{x}_4^{\text{int}(2)}$ deviates from the true $x_4^{\text{CF}(j)}$, it suffices to infer the correct signal $\phi_4(x_4^{\text{CF}(2)})$ regarding whether the fix suppresses the root cause at the target. Consequently, both interventions and CFs achieve near-perfect recall in our linear experiments, as reported in the main paper. However, for non-linear models, interventions surely emerge as a better choice.

## D.1   ADDITIONAL EXPERIMENTS ON LINEAR TOY DATASET

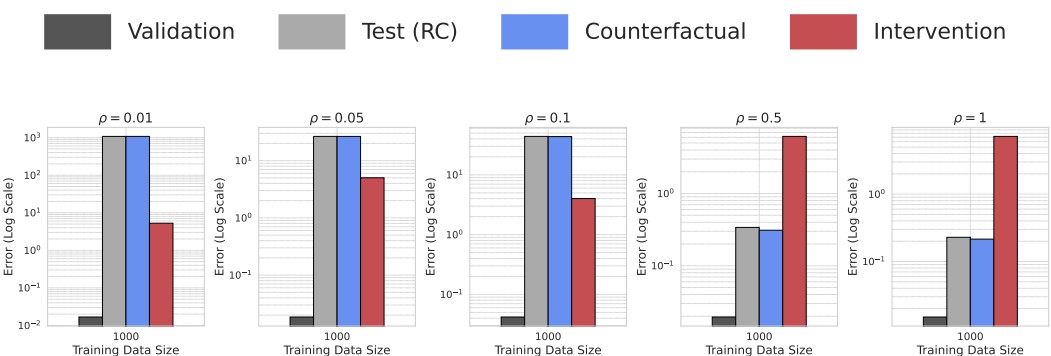

(a) Linear toy dataset with multi collinear features in the high training size regime and very high variance regime

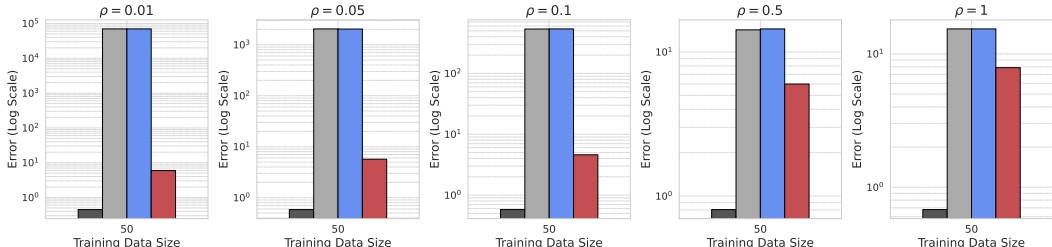

(b) Linear toy dataset with multi collinear features in the low training size regime and very high variance regime

Figure 8: Assessing the impact of variance of $\epsilon_i$ using a four variable toy dataset.

We extended the four-variable linear toy dataset above in the very high variance setting to examine scenarios where counterfactuals (CFs) and interventions perform differently. We considered low training size with 50 samples and high training size with 1000 samples to study the impact of such correlated features on the intervention and counterfactual errors. The input features, $X$, were expanded to six dimensions. The first three features, $X[:, 0:3]$, were sampled i.i.d. from a standard Gaussian distribution as before. The remaining three features were designed to be correlated with the first three as follows:

$$X[:,4] = X[:,1] + \mathcal{N}(0,\rho), \quad X[:,5] = X[:,2] + \mathcal{N}(0,\rho), \quad X[:,6] = X[:,3] + \mathcal{N}(0,\rho)$$

Here, $\rho$ takes values in $\{0.01, 0.1, 0.5, 1\}$. When $\rho = 0.01$, the fourth column is heavily correlated with the first, while for $\rho = 1$, the correlation coefficient is less pronounced.

The results shown in Fi. 8b, 8a reveal the following observations:

**For high training size and high variance settings:**
- When $\rho$ is small, interventions outperform CFs because multicollinearity induces high error in the abnormal root cause regime for CFs. Validation error remains close to zero across all $\rho$ settings, but CF test error blows up in the abnormal regime.

- As $\rho$ increases (e.g., $\rho > 0.5$), the effects of multicollinearity diminish, and CFs regain their dominance over interventions.

**For low training size across all $\rho$ values:**

- Interventions consistently outperform CFs. CF error escalates to as high as $10^3$, while interventions remain robust with their error bounded by the standard deviation of $\epsilon$.

**General Insights:** The central thesis of our work is that while there are specific scenarios where counterfactuals (CFs) outperform interventions—such as when the abducted values are accurately estimated and closely align with the oracle values (e.g., when $\hat{f}_i \approx f_i$)—these cases are exceptions rather than the norm. When consolidating the performance across the diverse SCMs examined in our study, interventions surely emerge as the superior approach for RCD compared to CFs.

# E  DESCRIPTION OF DATASETS

## E.1  PETSHOP

The PetShop application is a microservices-based pet adoption platform deployed on Amazon Web Services (AWS). It allows users to search for pets and complete adoption transactions. The system is built using multiple interconnected microservices, which include storage systems, publish-subscribe systems, load balancers, and other custom application based logic. These services were containerized and deployed using Kubernetes. The main focus of this dataset is to diagnose the root cause for anomalies that occur at a target node called PetSite – the front-end interface where users interact to browse for the pets displayed in the website. An anomaly is triggered at Petsite upon a violation of the service level objective (SLO), such as when the website's response time exceeds 200 microseconds.

**Causal Graph.** The service dependencies in the PetShop application are captured in a directed graph, with edges indicating the call relationships between services. This call graph is inverted by reversing the edge directions to obtain the underlying causal graph for this dataset. While the Oracle causal graph is available in PetShop, the Oracle SCM remains unavailable since the true structural equations underlying the Oracle SCM are unknown. Moreover, obtaining the exact Oracle SCM (i.e., the precise functions $f_i$) is infeasible for any real-world dataset, and rather only observed samples can be recorded. Previous studies leveraging PetShop, such as (Okati et al., 2024) have adopted a learning-based approach, approximating the Oracle SCM by training the structural equations on the collected finite samples. Our work also follows the same methodology.

**Node KPIs.** The target node, PetSite, has several ancestor nodes, including PetSearch ECS Fargate, petInfo DynamoDB Table, payforadoption ECS Fargate, lambdastatusupdater Lambda Function, and petlistadoptions ECS Fargate. Key Performance Indicators (KPIs) for these nodes were collected using Amazon CloudWatch, with system traces logged at 5-minute intervals. Metrics include the number of requests, and latency (both average and quantiles) for each microservice. Overall, the dataset includes 68 injected issues across five nodes, with each issue associated with a unique ground-truth root cause. Consequently, each test case in this dataset features a **unique** root cause.

**Root Cause Test cases.** The issues injected into the system span various types, including request overloads, memory leaks, CPU hogs, misconfigurations, and artificial delays. These issues affect different services, such as petInfo DynamoDB Table, payforadoption ECS Fargate, and lambdastatusupdater Lambda Function.  Each root cause test case is created by selecting a root cause node and then issuing an abnormally high number of requests through that node. This surge in traffic from the selected root cause node ultimately triggers an SLO violation at the Petsite node.  The dataset includes three types of root cause test cases: Low, High, and Temporal latency. On average, the request rates were 485 requests per second for Low latency, 690 for High latency, and 571 for Temporal latency cases. In the High and Temporal latency cases, the root cause nodes deviated significantly from the request patterns observed during training, allowing baseline methods such as traversal to perform well. However, for the more challenging Low latency cases, where the statistical discrepancy of the root cause nodes between the training regime and the abnormal root cause regime is less pronounced, IDI demonstrated the best performance.

The delays are parameterized with varying severities and affect different services, such as:

- PetSearch ECS Fargate: 500-2000ms delays for bunny search requests
- payforadoption ECS Fargate: 250-1000ms delays for all requests
- Misconfiguration errors: Affecting 1-10% of requests depending on the service

## E.2 SYNTHETIC DATASETS

We experiment with several synthetic datasets to benchmark the perfomance of the baselines against IDI. We desribe the generation of the causal graph first.

**Causal Graph.** We follow the graph generation procedure outlined in prior work (Budhathoki et al., 2022b). Given the total number of nodes $n$ and the number of root nodes $n_r$, the causal graph is constructed as follows: The nodes $X_1, X_2, \ldots, X_{n_r}$ are assigned as root nodes. For each internal node $X_k$ (where $k > n_r$), we randomly select its parent nodes from the set of preceding nodes $X_1, X_2, \ldots, X_{k-1}$. Each internal node has at most one parent, with a slight bias towards selecting a single parent. This sampling strategy helps to maintain a hierarchical structure and ensures the graph remains sufficiently deep. Once the parent nodes are chosen, directed edges are added from each parent to $X_k$. This process is repeated until all $n$ nodes are connected, forming a directed acyclic graph. Finally, we randomly choose a node that is at least 10 levels deep from a root node to serve as the target anomalous node. We show some example causal graphs in Fig. 14.

**Node KPIs.** Since cloud KPIs take on positive values – such as latency, number of requests, throughput, and availability (Hardt et al., 2024; Meng et al., 2020)—which are the commonly logged metrics, we ensured that all the nodes remain positive. To achieve this at root nodes, we sample $\epsilon_i$ from a uniform distribution $U[0, 1]$. For the internal nodes, we generate their observed values using one of the following three approaches:

- **Linear:** For a node $X_i$ with parents $\mathrm{Pa}_{X_i}$, we define the functional equation linearly with random coefficients as: $x_i = w_i^\top \mathrm{Pa}_{x_i} + \epsilon_i$ where $w_i \sim U[0.5, 2]$. These coefficients ensure that the KPI remains positive. We did not use $U[0, 1]$ for the weights because doing so caused deeper nodes in the graph to diminish, eventually reaching zero.
- **Non-Linear Invertible:** For non-linearity, we define each local mechanism $f_i$ as an additive noise three-layer ELU-activated MLP. We initialized the MLP weights using uniform distribution to ensure that even the internal nodes remain positive. Since many $X_i$ nodes have only one parent, we found that using four hidden nodes in the MLP was sufficient to generate non-trivial test cases. In this case, the equation becomes: $X_i = \mathrm{mlp}(\mathrm{Pa}_{X_i}) + \epsilon_i$ where $\epsilon_i \sim U[0, 1]$.
- **Non-Linear Non-Invertible:** In this setting, we use the same MLP architecture as in the previous case, but without additive noise. Here, each MLP $f_i$ receives both the parents $\mathrm{Pa}_{x_i}$ and the noise $\epsilon_i$ as inputs. Thus, the equation is: $X_i = \mathrm{mlp}(\mathrm{Pa}_{X_i}, \epsilon_i)$ with $\epsilon_i \sim U[0, 1]$. This configuration generates the most complex test cases in our experiments.

**Root Cause Test cases.** We consider two cases for the root cause test cases:

- **Unique:** In this case, we randomly sample a node from the ancestors of the target node.
- **Multiple:** Here, we randomly sample at most three nodes from the ancestors, ensuring that our assumption 1 holds. Specifically, no two root causes lie on the same path to the target. We experiment with test cases that satisfy this assumption and perform ablations to investigate its impact.

Suppose $\alpha^\star$ denotes the root cause node set. For $\alpha^\star$ to cause an anomaly at $x_n$, we apply a grid search over $\epsilon_{\alpha^\star}$ such that ($\epsilon_{\alpha^\star}$ they lead the SCM $\mathcal{S}$ to induce an anomaly at $x_n$. We start our search from 0 and increase them in steps of size 0.25 until the Z-score of the target node $\phi_n(x_n)$ hits the anomaly threshold 3.

## E.3 EXPERIMENTS UNDER ASSUMPTION 1 VIOLATIONS

In this experiment, we injected root causes at arbitrary nodes, resulting in Assumption 1 violations. The results are shown in Table 5. Both IDI and other Causal Anomaly methods face challenges in this scenario as they need parents of a root cause node to be usual. While `CF attribution` per-

| | Linear | | Non-Lin Inv | | Non-Lin Non-Inv | |
|---|---|---|---|---|---|---|
| | Top-1 | Top-3 | Top-1 | Top-3 | Top-1 | Top-3 |
| Random Walk (Yu et al., 2021) | 0.10 | 0.10 | 0.13 | 0.13 | 0.13 | 0.13 |
| Ranked Correlation (Hardt et al., 2024) | 0.00 | 0.00 | 0.00 | 0.00 | 0.00 | 0.00 |
| Traversal (Chen et al., 2014) | 0.00 | 0.40 | 0.03 | 0.27 | 0.00 | 0.27 |
| Smooth Traversal (Okati et al., 2024) | 0.23 | 0.50 | 0.17 | 0.47 | 0.30 | 0.43 |
| CIRCA (Li et al., 2022) | 0.13 | 0.13 | 0.27 | 0.27 | 0.13 | 0.13 |
| TOCA (Okati et al., 2024) | 0.07 | 0.07 | 0.03 | 0.03 | 0.00 | 0.00 |
| CF Attribution (Budhathoki et al., 2022b) | **0.83** | **0.97** | 0.33 | 0.57 | 0.07 | 0.23 |
| IDI | 0.57 | 0.57 | **0.53** | **0.60** | **0.40** | **0.53** |

Table 5: Experiment under Assumption 1 violations. A simple path to $X_n$ can features more than one root cause.

forms best in the linear setting, it struggles in other settings due to abduction errors being amplified by the presence of *multiple* root causes in the same path. For non-lin inv SCMs, IDI achieves the highest Recall, while in non-lin non-inv cases, it surpasses the CF method by $2\times$ Recall. Overall, IDI achieved the best method Recall even under assumption 1 violations.

# F  TIMING ANALYSIS

| | Method | PetShop | Syn Linear | Syn Non-Linear |
|---|---|---|---|---|
| Correlation | Random Walk (Yu et al., 2021) | 2.36 | 1.74 | 4.81 |
| | Ranked Correlation (Hardt et al., 2024) | 0.60 | 0.21 | 2.99 |
| | $\epsilon$-Diagnosis (Shan et al., 2019) | 2.11 | – | – |
| Causal Anomaly | Circa (Li et al., 2022) | 0.52 | 0.36 | 2.73 |
| | Traversal (Chen et al., 2014) | 0.27 | 0.24 | 1.05 |
| | Smooth Traversal (Okati et al., 2024) | 0.30 | 0.26 | 0.99 |
| Causal Fix | HRCD (Ikram et al., 2022) | 11.69 | – | – |
| | TOCA (Okati et al., 2024) | 1.96 | 0.95 | 9.16 |
| | CF Attribution (Budhathoki et al., 2022b) | 9.71 | 22.99 | 178.47 |
| Ours | IDI (CF) | 0.42 | 0.38 | 8.31 |
| | IDI | 0.37 | 1.29 | 9.62 |

Table 6: Running time (in seconds) for datasets with **unique** root causes. "–" indicates that the baseline was not consider for the corresponding dataset.

We present the running time required for predicting the unique root cause across all methods for one test case in Table 6. We show the results for the semi-synthetic PetShop dataset, as well as the Linear and Non-Linear versions of our synthetic datasets. Note that we omit the Non-Linear Non-Invertible cases because their running times were comparable to the Non-Linear Invertible cases. We make the following observations:

1. The Correlation and Causal Anomaly methods demonstrate the best performance in terms of running time.
2. Causal Fix approaches, on the contrary, are bottlenecked by the need to learn the Structural Causal Model (SCM). Learning the SCM involves fitting a lightweight regression model $\hat{f}_i : \mathrm{Pa}_{X_i} \mapsto X_i$ for each node $i$. Recall that these models are lightweight because they only need to regress the parent covariates of the nodes. The Linear methods incur less time compared to the Non-Linear ones, as they can be learned using closed-form expressions, whereas Non-Linear methods require gradient descent-based training.
3. For predicting the unique root cause, both IDI (CF) and IDI do not require Shapley value computations, allowing them to run in significantly less time.
4. The baseline CF Attribution method, however, performs Shapley analysis across all nodes, even for the unique root cause, making it the worst-performing method in terms of running time.

Table 7 presents the results for the running time required to predict multiple root causes. Unlike the unique root cause, our method IDI and its CF ablation IDI (CF) require Shapley analysis in this setting. However, Shapley values are computed only for the subset of nodes in $\mathcal{R}_{\mathrm{cand}}$, identified after the first step of our algorithm (the Anomaly condition). We make the following observations:

| | Method | Syn Linear | Syn Non-Lin Inv | Syn Non-Lin Non-Inv |
|---|---|---|---|---|
| Corr. | Random Walk (Yu et al., 2021) | 8.63 | 6.73 | 9.1 |
| | Ranked Correlation (Hardt et al., 2024) | 0.26 | 1.54 | 1.9 |
| Causal Anomaly | Circa (Li et al., 2022) | 0.38 | 2.34 | 2.23 |
| | Traversal (Chen et al., 2014) | 0.22 | 2.26 | 1.99 |
| | Smooth Traversal (Okati et al., 2024) | 0.26 | 2.37 | 1.95 |
| Causal Fix | TOCA (Okati et al., 2024) | 1.3 | 11.66 | 13.58 |
| | CF Attribution (Budhathoki et al., 2022b) | 23.48 | 120.61 | 190.08 |
| Ours | IDI (CF) | 7.08 | 40.12 | 73.41 |
| | IDI | 8.2 | 42.6 | 76.24 |

Table 7: Running time (in seconds) for datasets with **multiple** root causes.

1. The running time for the random walk-based approach increases due to the presence of multiple anomalous paths leading to the target node.
2. All other baseline approaches exhibit running times comparable to those observed in the unique root cause test cases.
3. The running times for IDI and IDI (CF) increase because of the additional Shapley computations. However, this increase is significantly smaller compared to CF Attribution, as the former computes Shapley values over a subset of nodes, while the latter evaluates them across all nodes.

| Dataset | CF Attrib. | IDI (Ours) |
|---|---|---|
| Linear | $36.4 \pm 3.6$ | $6.0 \pm 1.6$ |
| Non-Linear Invertible | $35.8 \pm 4.2$ | $6.0 \pm 1.9$ |
| Non-Linear Non-Invertible | $37.1 \pm 3.8$ | $6.2 \pm 1.9$ |

Table 8: Number of nodes considered for Shapley Analysis: We report the mean $\pm$ standard deviation computed across all the test cases.

We report the number of nodes involved in computing the Shapley values in Table 8. Since Shapley value computations are NP-Hard, in practice, Monte Carlo simulations are commonly used to approximate these values by sampling permutations of the nodes. In our work, we sampled 500 permutations for all methods to ensure tractability. Table 8 presents the mean number of nodes involved, along with the standard deviation across all test cases. Overall, we observe a $6\times$ reduction in the number of nodes for IDI compared to the CF Attribution baseline. Notably, if exact computation of Shapley values were performed, this reduction factor would be even more significant.

# G   EXPERIMENTS WITH HIGH VARIANCE OF $\epsilon_i$

In this section, we experiment with different sampling distributions for $\epsilon_i$ to evaluate their impact on interventional and counterfactual estimates. We begin with a simple four-variable toy example.

## G.1   SYNTHETIC SETTING

In this subsection, we evaluate the impact of $\epsilon_i$ variance on other datasets used in our study.

**PetShop.** For the real-world PetShop dataset, the exogenous variables $\epsilon$ are latent, preventing us from characterizing or controlling their variance. So we cannot experiment with this dataset.

All the synthetic experiments outlined below use 100 i.i.d. training samples to learn the SCM $\widehat{\mathcal{S}}$.

**Linear SCM.** In the main paper, we conducted experiments with each $\epsilon_i$ sampled from $U[0, 1]$. Here, we explore broader distributions by sampling from $U[0, b]$, with $b \in \{0.5, 2, 3, 5\}$. To reduce clutter, we focus on the best-performing baselines from the main results. Figure 9 presents results for the Linear SCM. The left panel corresponds to unique root cause test cases, while the right panel shows multiple root cause test cases, with Recall@1 on the Y-axis. As expected, in the linear setting, both interventional and counterfactual variants of IDI achieve Recall of 1 across all variance settings, with the CF Attribution baseline standing out as a strong competitor.

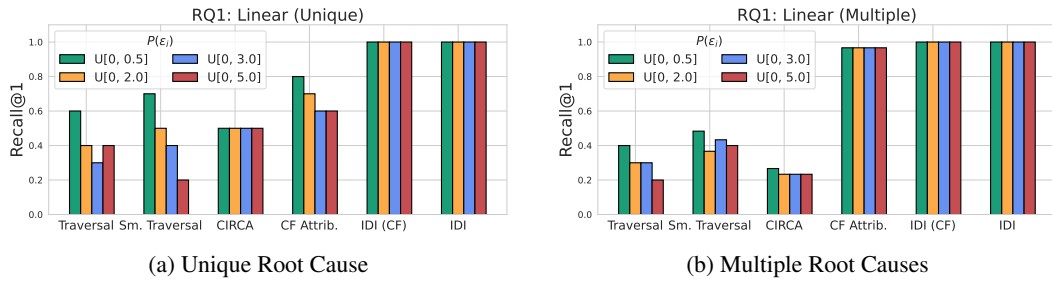

(a) Unique Root Cause

(b) Multiple Root Causes

Figure 9: Linear Oracle SCM

**Non-Linear Invertible.** Figure 10 presents results for this setting. For unique root cause test cases, IDI slightly outperforms the IDI (CF) variant of our approach. Among the baselines, CIRCA and Smooth Traversal emerge as strong competitors, while CF Attribution performs poorly at high variance. For multiple root causes, IDI falls slightly short of IDI (CF) in high variance scenarios, likely due to the overfitting or unstable training of the SCM $\widehat{\mathcal{S}}$. We infact observed high validation errors during training for high variances $\epsilon_i$s.

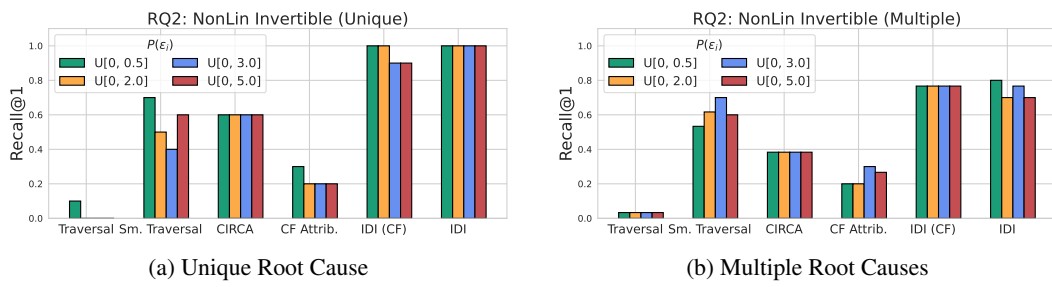

(a) Unique Root Cause

(b) Multiple Root Causes

Figure 10: Non Linear Invertible Oracle SCM

**Non-Linear Non-Invertible.** This setting is the most challenging among all datasets because $\epsilon_i$ is not identifiable, causing CF approaches to struggle. As shown in Fig. 11, both CF Attribution and our IDI (CF) variant perform poorly, often achieving near-zero recall at high variance. While IDI also experiences performance drops compared to previous datasets, it remains comparatively robust when compared against the baselines. IDI stands out as the best approach across all variance settings in this dataset.

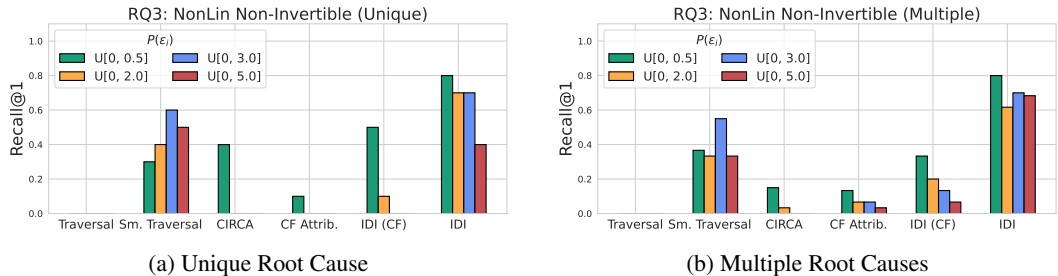

(a) Unique Root Cause

(b) Multiple Root Causes

Figure 11: Non Linear Non Invertible Oracle SCM

# H  SENSITIVITY ANALYSIS OF THE ANOMALY THRESHOLD

We conducted this experiment on the PetShop dataset. In our main paper, we used a default anomaly threshold of 5. In this section, we assess the impact of baselines that implement the anomaly condition and IDI under varying anomaly thresholds, $\tau_i$. For Z-Score, the threshold determines how many

standard deviations a sample must deviate from the mean to be considered anomalous. We experimented with thresholds ranging from 2 to 7 and report the results in Table 9. Overall, we observed that IDI and the other baselines remained robust across different threshold choices, with IDI only showing performance degradation at a threshold of 2. We acknowledge that tuning the threshold is important in practice. However to tune it, we abnormal root cause test samples during training, since most samples in $D_{\text{trn}}$ are non-anomalous. In the absence of such abnormal test samples, specifying this hyperparameter involves domain expertise.

| | Recall@1 | | | | | | Recall@3 | | | | | |
|---|---|---|---|---|---|---|---|---|---|---|---|---|
| | 2 | 3 | 4 | 5 | 6 | 7 | 2 | 3 | 4 | 5 | 6 | 7 |
| Traversal | 0.73 | 0.80 | 0.87 | 0.90 | 0.87 | 0.80 | 0.73 | 0.83 | 0.87 | 0.90 | 0.87 | 0.80 |
| Smooth Traversal | 0.30 | 0.48 | 0.30 | 0.30 | 0.30 | 0.30 | 0.73 | 0.73 | 0.70 | 0.67 | 0.73 | 0.67 |
| IDI (CF) | **0.80** | 0.80 | 0.80 | 0.80 | 0.80 | 0.80 | 0.90 | 0.90 | 0.90 | 0.90 | 0.90 | 0.90 |
| IDI | 0.73 | **0.93** | **0.90** | **0.93** | **0.93** | **0.93** | **0.93** | **0.93** | **0.93** | **0.93** | **0.93** | **0.93** |

Table 9: Results under variation of the anomaly detection threshold in the PetShop dataset. We show both Recall1@1 and Recall@3.

# I  SYNTHETIC CAUSAL GRAPHS

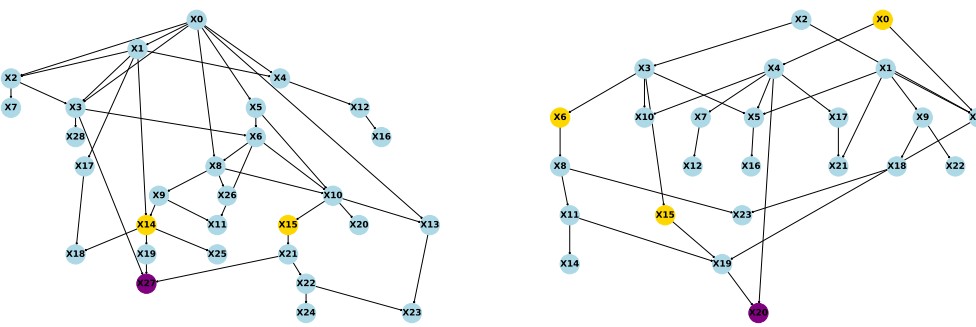

Figure 12: An example Synthetic test case     Figure 13: An example Synthetic test case

Figure 14: Random graphs sampled for our Synthetic Experiments

We show two instances of random graphs in Fig. 14. The purple node is the anomaly for which we need to find the root cause. The ground truth root cause nodes are shown in yellow. We typically observed that all nodes that are descendants of the yellow nodes also tend to exhibit anomalous behavior.

