# OpenReview forum: "Robust Root Cause Diagnosis using In-Distribution Interventions"
_ICLR.cc/2025/Conference — ICLR 2025 Poster_

### Official Review · Reviewer_ERJh · 2024-10-30

**Soundness:** 3
**Presentation:** 3
**Contribution:** 3
**Rating:** 8
**Confidence:** 3

**Summary:**

This paper introduces the In-Distribution Interventions (IDI) method to address the problem of RCD. The authors highlight two critical conditions in the RCD problem: the anomaly condition and the fix condition. The authors demonstrate that intervention-based approaches outperform counterfactual reasoning when tackling RCD problems, particularly by avoiding OOD errors commonly associated with counterfactual estimation. The proposed RCD algorithm uses an intervention-based approach to handle both conditions simultaneously. The evaluation results show the effectiveness of the IDI algorithm and its robustness in handling complex anomaly environments.

**Strengths:**

Strengths:
- This paper proposes a novel in-distribution intervention algorithm to address the RCD problem, which simultaneously considers both the fix and anomaly conditions in RCD, demonstrating strong robustness in complex failure scenarios.
- A well-supported theoretical proof. The authors demonstrate the advantages of the intervention method over counterfactual reasoning when dealing with OOD scenarios and apply this method to the fixed condition in the IDI method.
- Based on the evaluation results, IDI demonstrates better performance and robustness than the existing baseline algorithms across different datasets.
- Good reproducibility. The authors have open-sourced the code, data, and detailed instructions.

**Weaknesses:**

Weaknesses:
- The OOD problem is not clearly presented. The OOD problem in RCD is an important motivation for the paper. However, I could not find a formal definition of OOD or in-distribution in RCD in the paper. The authors should provide a formal definition of OOD in the context of RCD early in the paper, e.g., in the introduction or background. Additionally, the authors should include an illustrative diagram contrasting OOD and in-distribution scenarios to help clarify this concept for readers.
- More details about the construction of the SCM are necessary. In the paper, the authors directly apply an Oracle SCM rather than constructing it in an automated way, which is uncommon in existing RCD algorithms. The authors should carefully explain how the Oracle SCM was obtained and discuss whether this method can be generalized.
- The description of CIRCA may have some minor issues. In CIRCA, the authors use the Descendant Adjustment technique to handle the anomaly scores of downstream nodes in the CBN from the root cause. Therefore, Descendant Adjustment can mitigate the issue with CIRCA, as mentioned in line 276 of the paper. The authors should describe this technique in the paper.
- The authors need to provide a more precise overview of the IDI algorithm. Currently, the proof in Section 4 takes up too much space, while the description of the IDI algorithm in Section 5 needs to be more brief and mainly presented in text. The authors should include an overview figure in Section 5 to help readers understand the overall workflow of the IDI algorithm.
- Lack of explanation of the Petshop dataset. In Table 1, the authors evaluate the results across three categories: Low, High, and Temporal. However, the paper needs to explain why these categories are separated for comparison or what observation in each category is meant to reflect. The authors should briefly explain these categories in the Petshop dataset, including why they are relevant to RCD and what insights they provide about the performance of different methods.
- In Section 6.3, cases of Assumption 1 violations are evaluated. However, the paper does not explain the motivation for this evaluation or whether such violations occur in real-world scenarios. The authors should explain the real-world relevance of these violations and how common they might be in practice.
- Lack of discussion on some existing counterfactual reasoning methods. Sage [1] proposed using CVAE to generate counterfactuals for RCD. The authors should discuss and compare this method in the paper.
- Lack of case studies on IDI with OOD data. Although the experiments show promising results, the authors should at least present a case study where IDI is applied for RCD in an OOD scenario (e.g., step-by-step results of applying IDI to the OOD cases with intermediate outputs, a comparison of how IDI performs on OOD cases v.s. other methods) to validate its effectiveness on OOD data.
- Shapley values are used for multiple root causes, possibly introducing high time complexity. The paper does not discuss the impact of this on scalability. The authors should discuss the computational complexity of their Shapley value approach and its implications for scalability in large-scale RCD scenarios, provide empirical runtime analysis, or discuss potential optimizations for improving efficiency.

[1] Gan, Yu, Mingyu Liang, Sundar Dev, David Lo, and Christina Delimitrou. "Sage: practical and scalable ML-driven performance debugging in microservices." In Proceedings of the 26th ACM International Conference on Architectural Support for Programming Languages and Operating Systems, pp. 135-151. 2021.

**Questions:**

- Can you provide a clear and formal definition of OOD in the context of the RCD problem?
- How will the performance of IDI be affected if an Oracle SCM cannot be obtained? How can this algorithm be applied in real-world scenarios where an Oracle SCM is unavailable?
- Will using the Shapley value reduce algorithm efficiency in large-scale RCD scenarios?

---

> ### Author Response · Authors · 2024-11-21
> **Response 1/n**
>
> Thank you for your kind words and appreciation of our work! We are delighted that our code has been helpful.
>
> > 1 (a) OOD Problem
>
> Any input lying outside the training distribution is generally considered OOD. Since characterizing the training probability distribution is challenging, we can loosely define a node $X_i$ as taking an OOD value when its Z-Score is significantly high. The Z-Score measures how many standard deviations a sample lies from its mean. For example, the Petshop dataset by default declares values that are five standard deviations away from the mean as OOD.
>
> > 1 (b) Illustrative Figure for In-Distribution and OOD Scenarios
>
> We request the reviewer to refer to `Figure 1` in the paper. We have updated the caption to explicitly mention the OOD scenario. Additionally, we have introduced a new `Figure 2` in `Section 3` that illustrates the RCD pipeline of IDI. We would be happy to incorporate further suggestions and answer any follow-up questions.
>
> > Q2: Details of the Oracle SCM
>
> Please note that IDI only assumes knowledge of the dependency graph aka causal graph between nodes, and not the SCM.
>
> For the Petshop dataset, we do not know the Oracle SCM. However, for our synthetic datasets, we constructed several types of Oracle SCMs. Detailed explanations of how these SCMs were designed are included in` Appendix C.2` in the revised manuscript.
>
> IDI trains the SCM using samples from the training data. Training the SCM involves fitting $n$ regression functions, one for each node. The second block in our new `Figure 2` illustrates the SCM training process.
>
> > Q3: CIRCA description has minor issues
>
> The authors of CIRCA made a *very* valuable observation in their paper that in RCD test cases, a root cause node drives its descendants to take on anomalous or OOD values. This insight led them to develop the descendant adjustment technique as an effective way to prevent predicting descendants of the root cause node as root causes. We leveraged this observation to support our claim that this phenomenon results in large $tvd$ terms in the CF bound in our theory. Therefore, the contributions of CIRCA complement our work. However, as noted by reviewer `7kui`, this detail might be considered a distraction. In response, we have removed this comment from the remark in our revised manuscript.
>
>
> > Q4: Precise Overview of Algorithm and Pipeline Figure
>
> In the revised draft, we have significantly condensed the methods section. Additionally, we included `Figure 2` to provide a visual illustration of the IDI pipeline. We request the reviewer to take a look and share further feedback.
>
>
> > Q5 (a) Petshop details
>
> We have added these details in `Appendix C.1`
>
> > Q5 (b) Why are three test cases different?
>
> We reported low, high, and temporal latency separately because the nature of anomalies injected for each category is distinct (refer `Sec. C.1`). Low latency test cases involved fewer requests compared to high latency and temporal ones, making root cause identification more challenging for the low latency cases. Causal anomaly methods, such as traversal, achieved perfect recall on temporal test cases that exhibit more prominent anomaly behavior. To emphasize the efficacy of IDI in challenging low latency test cases, we presented the results separately.
>
>
> > Q6 Violations of Assumption 1
>
> Our method, IDI, relies on this assumption to correctly identify root causes in multiple root cause test cases. Therefore, we evaluated its effectiveness when the assumption is violated. If the reviewer feels that this experiment is a distraction, we are open to moving it to the appendix.
>
>
> > Q7: SAGE
>
> This paper is highly relevant to our work, and we sincerely thank the reviewer for pointing it out. SAGE is a counterfactual causal fix approach where the authors train CVAEs for each node during SCM training. To assess the fix condition, the method infers the posterior distribution over the exogenous variables $\epsilon$s (note that SAGE uses the notation $Z$ to denote the $\epsilon$) by conditioning the CVAE encoder on the root cause test features. As noted by CIRCA and other prior works, test samples in RCD often contain many anomalous values. Consequently, SAGE also incurs OOD evaluations of the SCM, unlike our approach, IDI.
> Unfortunately, we could not find the code and were unable to include a comparison with this baseline. However, we have cited and discussed it at `line 122` in our related work section.

---

> > ### Author Response · Authors · 2024-11-21
> > **Response 2/2**
> >
> > > Q8: Step-by-step analysis of IDI
> >
> > Thank you for this question. To compare the pros and cons of taking an interventional approach as done by IDI versus commonly used counterfactual estimates, we conducted an in-depth experimental analysis with a simple four-variable toy dataset.
> >
> >
> > We have updated the anonymized repository to include a notebook (https://anonymous.4open.science/r/petshop-BB8A/limitation.ipynb) that provides a detailed explanation through code. We hope the reviewer finds this helpful.
> > In summary, by experimenting with several different synthetic Oracle SCMs, we established the types of datasets that favor IDI over counterfactual estimates when applied to OOD anomalous test cases. We kindly request the reviewer to refer to `Sec. E` in the appendix for details.
> >
> >
> >
> > > Q9: Timing Analysis
> >
> > We have conducted a detailed timing analysis during rebuttal. For a brief overview, please refer to Q2 of reviewer `hq7T`. For a more comprehensive analysis, please see `Appendix D`.
> >
> > # Questions
> >
> > > Q1 definition of OOD in the context of the RCD
> >
> > We have enhanced the clarity in `Fig. 1`. We hope that this addresses the reviewer’s question.
> >
> > > Q2 If an Oracle SCM cannot be obtained?
> >
> > IDI does not rely on the availability of the Oracle SCM. Instead, it learns the SCM from the training samples. In our toy experiment in Appendix `Sec. E`, we demonstrate the trade-offs between training size and IDI performance across different Oracle SCM settings.
> >
> > Our Non-Linear Non-Identifiable Oracle SCM scenario represents a case where the Oracle SCM cannot be identified even with infinite training samples. Our experiments showed that in such cases, our intervention-based IDI approach outperforms counterfactual (CF) based baselines by a significant margin.
> >
> > In real-world applications, such as the Petshop dataset, we do not have the exact characteristics of the Oracle SCM. Hence, we can only rely on empirical performance where we observed that IDI performs best. Further, the consolidated performance across all the datasets considered in our work suggests that IDI is a more robust approach across different Oracle SCM configurations.
> >
> > > Q3 Shapley values and efficiency
> >
> > We compared the overhead of using Shapley values for multiple root causes with that of the CF-based attribution baseline that also uses Shapley values. Since IDI computes Shapley values only for candidate nodes in $R_{cand}$, its complexity is significantly lower compared to the baseline approach, which computes Shapley values for all nodes in the dataset. Therefore, the computational cost should not pose a significant bottleneck in practice. Please see below the number of nodes considered by the two methods.
> >
> > | **Dataset**                  | **CF Attribution** | **Ours**       |
> > |------------------------------|--------------------|----------------|
> > | **Linear**                   | 36.4 ± 3.6         | 6.0 ± 1.6      |
> > | **Non-Linear Invertible**    | 35.8 ± 4.2         | 6.0 ± 1.9      |
> > | **Non-Linear Non-Invertible**| 37.1 ± 3.8         | 6.2 ± 1.9      |
> >
> > **Observations**
> >  - Shapley value computations are NP-Hard and typically require Monte Carlo approximations. To ensure tractability, we used 500 permutations for all methods.
> > -  IDI achieves a **6× reduction** in the number of nodes analyzed compared to CF Attribution, significantly lowering the computational cost. The reduction would be even more pronounced with exact Shapley computations.

---

> > > ### Author Response · Authors · 2024-11-24
> > > **Requesting feedback on our rebuttal**
> > >
> > > Dear Reviewer ERJh,
> > >
> > > Thank you for your constructive feedback on our paper. With the ICLR public discussion phase ending shortly, we wanted to confirm whether our responses have sufficiently addressed your concerns. If there are any remaining issues, we would be happy to provide additional clarifications.
> > >
> > > Thank you very much for your time!

---

### Official Review · Reviewer_9ck8 · 2024-11-03

**Soundness:** 3
**Presentation:** 2
**Contribution:** 3
**Rating:** 6
**Confidence:** 2

**Summary:**

This paper proposes an algorithm for diagnosing the root causes of anomalies in cloud-based systems. It addresses limitations in traditional counterfactual-based approaches by using in-distribution interventions on structural causal models to identify root causes with enhanced robustness and accuracy. By avoiding out-of-distribution issues common in counterfactuals, it achieves better performance in terms of robustness and root cause accuracy, as validated through experiments on synthetic and benchmark datasets.

**Strengths:**

- The paper provides a solid theoretical foundation for IDI, including error bounds and conditions where IDI surpasses counterfactual methods.

- Experimental results consistently show that IDI outperforms baselines, highlighting its effectiveness and robustness in root cause diagnosis.

**Weaknesses:**

- The presentation is complex and may be challenging to follow, particularly in Section 3, due to the heavy use of notation. Adding examples to clarify key concepts would improve readability.

- While IDI demonstrates strong performance on synthetic and benchmark datasets, further validation in diverse, real-world industrial systems would strengthen its practical applicability.

- The focus is on accuracy, but runtime and latency evaluations are limited. A discussion of computational overhead and potential optimizations for large-scale deployment would increase its relevance.

**Questions:**

- Could counterfactual methods still be preferred over IDI in certain cases or types of systems? If so, can you give insights on how to choose between these approaches?

- Is IDI sensitive to variations in anomaly detection thresholds?

---

> ### Author Response · Authors · 2024-11-21
> **Response**
>
> Thanks so much for appreciating the theoretical and experimental contributions of our work.
>
> > Q1 Complex presentation
>
> We have revised the paper considering your feedback and the suggestions from other reviewers. Specifically, we have:
>
> - Explicitly highlighted the OOD challenge in `Fig. 1`
> - Trimmed down `Sec. 5`.
> - Added the pipeline illustration in `Fig. 2`.
> - Included a simple toy experiment in `Appendix E` to improve understanding.  We also updated the anonymized repository with a detailed explanatory notebook: https://anonymous.4open.science/r/petshop-BB8A/limitation.ipynb.
> - Explicitly highlighted the limitations in `Sec. 7`.
>
> We hope these additions address the concerns. We are open to further feedback to enhance the paper.
>
> > Q2 Validation in diverse real-world datasets4
>
> We could not perform this experiment because for real-world industrial datasets because their dependency graphs are usually considered proprietary. Further, prior works like BARO (https://arxiv.org/pdf/2405.09330) have noted that causal approaches like CIRCA fail due to the improper recovery of the causal graphs. So to focus on just the root cause prediction efficacy, we solely experimented with benchmarks containing dependency graphs.
>
> > Q3 Timing Analysis
>
> We request the reviewer to refer to Q2 of reviewer `hq7T`. We have added a very detailed timing analysis in the `Sec. D` in the Appendix of the revised manuscript.
>
> > Q4 When is CF method preferred over IDI
>
> As a short answer, when the underlying true SCM allows for identifiability of the exogenous variables. In the limit of training size, when the estimated structural equations in the learned SCM become perfect, the estimates provided by the counterfactual methods converge to the true counterfactual values. On the contrary, the error using interventional estimates considered by IDI would plateau at the standard deviation of the exogenous variables. We request the reviewer to refer to `Appendix Sec. E`, or the code in our notebook for a detailed explanation. We have added this as an explicit limitation in `Sec. 7` in our revised manuscript. Thanks very much for this question.
>
> > Q5 Sensitivity to anomaly thresholds
>
> We have added this experiment in `Appendix F`. We performed the sensitivity analysis on the Petshop dataset. Our observations include:
>
> - **Robustness Across Thresholds**: IDI and other baselines mostly demonstrated robustness across a wide range of anomaly thresholds from 2 to 7.
> - **Degradation at Low Thresholds**: IDI showed a slight performance degradation only at the lowest threshold of 2, highlighting sensitivity to overly lenient anomaly definitions.
> - **Tuning Challenges**: Determining the optimal threshold requires abnormal root cause test samples during training, which are often unavailable. In such cases, domain expertise is crucial for specifying this hyperparameter.

---

> > ### Author Response · Authors · 2024-11-24
> > **Requesting feedback on our rebuttal**
> >
> > Dear Reviewer 9ck8,
> >
> > Thank you for your constructive feedback on our paper. With the ICLR public discussion phase ending shortly, we wanted to confirm whether our responses have sufficiently addressed your concerns. If there are any remaining issues, we would be happy to provide additional clarifications.
> >
> > Thank you very much for your time!

---

> > > ### Author Response · Authors · 2024-12-02
> > > **Gentle Reminder**
> > >
> > > Dear Reviewer 9ck8,
> > >
> > > Thank you once again for your insightful feedback. We greatly appreciate the time and effort you have devoted to reviewing our work.
> > >
> > > As the ICLR public discussion phase nears its conclusion shortly, we wanted to ensure that our responses have sufficiently addressed your concerns. If there are any remaining questions or areas requiring further clarification, please let us know—we would be happy to provide additional details.

---

> > > > ### Comment · Reviewer_9ck8 · 2024-12-02
> > > >
> > > > Thank you for your response. I have increased my score accordingly.

---

### Official Review · Reviewer_7kui · 2024-11-03

**Soundness:** 3
**Presentation:** 1
**Contribution:** 2
**Rating:** 6
**Confidence:** 3

**Summary:**

The paper proposes an approach: IDI that identifies root causes based on two criteria: (1) the node exhibits anomalous behavior (anomalous condition), and (2) if the node had normal values, the target node would not be anomalous (fix condition). It contrasts this approach with the pre-existing methods that use counterfactuals from Structural Causal Models (trained on historical data)—often unreliable for rare anomalies (as they are OOD). IDI relies on interventional estimates within the training distribution (in reality the validation distribution).

**Strengths:**

The three strengths of the paper are:
1. Whatever is presented in the paper is technically sound. Although I believe some clarifications are needed and the contribution is not enough (i.e., more content can be added).
2. **Main Strength:** The experiment section is very impressive. It is thorough and the ablation studies are performed well.
3. The main research question addressed in this paper i.e., counterfactual estimates can push SCMs to OOD regions, and sampling from in-distribution to intervene to circumvent this issue, is an important contribution to the community. However, originality of this idea is unclear.
4. Monitoring cloud KPIs for anomalies and root cause diagnosis in them is an important application. Showing the experiments on "Petshop" is impressive. In a way the paper is written to cater the root cause diagnosis needs of Petshop. However, the authors need to provide some insights on cloud KPIs either in Introduction or Preliminary to make the paper even stronger for engineers.

**Weaknesses:**

The methodological contribution of the paper is a major weakness. There are some weaknesses in the experiment section as well.
1. Sampling the latent exogenous variable from the (validation set) distribution as opposed to inverting the function $f_i$ (abduction) is important but a minor methodological tweak that might not warrant a full research paper. It is also unclear if "in-distribution intervention" approach is novel or has been proposed in the literature. The authors must show an analogy of the abduction step as a probability distribution. And then they should show how sampling from that distribution is different from the proposed approach.

2. Most parts of the papers are pre-existing knowledge except the two theorems. Even the difference between the two theorems is the abduction step, which is intuitive from the definitions of counterfactual vs intervention (equation 4, 5, 6).

3. Section 5 is over-explained (especially for the unique root cause case). The definitions of "anomaly condition" and "fix condition" is well established before that section. The only contribution I see in section 5 (i.e., the paper's approach) is "Assumption 1" and "Multiple root causes explained using Shapley values".

4. "Low variance of exogenous variables $\epsilon_i$ is the most important assumption of this paper. It should have been reflected in the title, and the abstract. Furthermore, the variance of $\epsilon_i$s are not mentioned for the experiments, thus not providing transparency if the experiments satisfied this assumption.

5. No experiment is done for "high variance" of exogenous variable. Therefore a reader cannot get insights if counterfactual method is better for such cases (as opposed to IDI). Please conduct a study to compare IDI and counterfactual methods for a range of variances (of latent exogenous variable) -- clearly showing the boundary cases where counterfactual outperforms IDI.

6. In table 2 and table 4, a lot of the recall values for the proposed IDI approach seems 1.00, which raises questions on the data itself. The authors must provide insight if the data is biased towards their methodology. Just mentioning IDI outperforms the baseline is not enough, since the results are too good to be true. Please do the following -- (1) provide details on the data generation process including the randomness in the graph generation, (2) do multiple replications of the study and report the standard deviation of the recall values. If the authors have a reason to believe that the dataset might be biased for their proposed approach, they must provide reasons and mention the experimental limitations explicitly.

7. The authors must provide examples or case study to show that their assumption of low variance of latent exogenous variable is true. In page 2, authors mentioned that they are "typically small in practice" without any references, or examples. This is not acceptable for a technical publication. The authors are required to provide specific examples and/or case studies to support their claim of low variance of exogenous variables. The authors must also show how these claims are true in the experimental dataset (both synthetic and Petshop).

8. The paper is too verbose. The authors repeat the two conditions (anomaly and fix) over and over again, in multiple sections without new content. I suggest the authors to reduce the length of the current version of the paper. Add a few explanations to weakness pointed out be me and other reviewers. The authors need to understand that SCMs and causality might be jargon for a lot of readers. They need to explain everything using simpler notions of either probability or graphs like Figure 1 or Figure 2.

9. The paper's formatting is not acceptable. Page 5, Page 6 mentions theorem numbers are 5 and 6 and in supplementary they are written as theorems 10, 11 respectively. Similarly Theorem 6 is referring to Theorem 10, which is not a real thing in the main text of the paper. Similarly, in page 4, under "training setup", the symbol $D^{trn}$ is used which is not defined at all. The paper uses a lot of abbreviations and do not expand it. Some abbreviations like IDI are used even in abstract. These are incorrect formatting. I will assume the authors can make changes in rebuttal phase to these issues.

10. In section 3.3., for "1. Anomaly Condition", the authors write "prior methods....". Without references, such statements must be avoided and in my honest opinion unacceptable for a technical manuscript.

11. Remark on page 6 is not comprehensive. The authors must remove the "CIRCA" reference and explain only their method. Otherwise, it digresses from the context.

12. The authors mentioned in page 7 that their work uses "Z-score". Please provide some preliminary knowledge on Z-score and highlight where is it used ?

**Questions:**

1. Why did the authors used the phrase "auto-regressively" at multiple places in the paper ? Is the data assumed to be time series ? Is the author referring to topological order as "auto-regressive" ?

2. In page 9, for non lin non-invertible SCMs, the authors wrote "but without additive noise". Is non-additive noise the only way to induce non-invertibility in SCMs ?

3. In page 7, under multiple root causes it is mentioned "However, one problem is that applying fixes to any subset $\alpha$ of ....". This statement is not clear. This statement is the main reason behind using Shapley values and also the primary contribution of your multiple root cause subsection. Please rephrase this statement and explain it in simpler words.

4. I understand that literature uses the phrase "disrupted local causal mechanism" and/or "abnormal $\epsilon_j$ for "Anomaly Condition". How are they all the same thing? Is it because under anomalous $x_j$, $\epsilon_j$ is OOD ? The authors must provide explanation to this in the paper.

5. **Important Question:** If I take the sample mean of $\tilde{\epsilon_{j+1}}$ for a large enough sample size, will it or will it not converge to the $\hat{\epsilon}_{j+1}$ ? -- equation 4, 5, 6

6. In page 9, in the paragraph mentioning "Generating the Oracle SCM $\mathcal{S}$", it is mentioned that the "Cloud KPIs are usually positive". Why ? If thats a fact, give references. If thats an assumption, mention it explicitly.

I will wait for the rebuttal. Please address both weaknesses and questions.

---

> ### Author Response · Authors · 2024-11-21
> **Response 1/n**
>
> Thank you so much for appreciating our experiments and for providing a thorough and rigorous review of our work. We enjoyed writing the rebuttal.
>
> > Writing enhancement
>
> We included examples for cloud KPIs in the intro. Thanks for this suggestion.
>
> # Weaknesses
>
> We grouped answers for several questions here.
>
> > Q1,2: Novelty and Pre-existing Knowledge
>
> We appreciate the reviewer’s acknowledgment of our theoretical contributions. In addition to the theoretical analysis, our proposed algorithm’s  novelty lies in how we sequence the evaluation of the anomaly and fix conditions so as to avoid OOD estimates.  The algorithm may seem simple in retrospect, but we explored several variants before finally arriving at the IDI algorithm, and precisely characterizing the condition under which it works.  We are aware of no prior work on RCA that ensures robustness by avoiding OOD probing in the setting of multiple root causes and single cause per path.  The robustness of IDI manifests in the consistently better results we obtain under diverse settings of complexity of SCM, variance of exogenous noise, and training set size. We request the reviewer to refer to `Sec. E` in the Appendix added during rebuttal for a fine-grained empirical analysis. Now, we answer in detail:
>
> **Pre-existing Knowledge:**  It is well-known that estimating counterfactuals (CFs) from observational samples often encounter impossibility results. In our setup, we assume the causal graph is available along with observational data, enabling CF estimation. Prior methods like DeepSCM [1] and Recourse under Imperfect Causal Knowledge [2] have demonstrated success in such scenarios. Hence, literature in Root Cause Diagnosis (RCD) widely regards counterfactual estimation as the correct approach to identify root causes [refer CF Attribution, [3]].
>
> **Novelty:**
> Given that CF-based methods dominate the causal RCD literature, our findings are significant and can shift the field towards intervention-based RCD methods. The only work we found that acknowledges this OOD challenge is TOCA. Our work complements TOCA by providing a more detailed analysis, attributing the issue to the abduction step (lines 79–85). However, TOCA’s procedure to assess the fix condition performed poorly in our experiments—a limitation also noted by its authors.  Our key novelty thus lies in designing an RCD algorithm that works exclusively with interventions, which is challenging given RCD's sensitivity to OOD estimation. Even algorithms that appear similar (e.g., IDI, TOCA both using interventions) can exhibit vastly different performance, as our results demonstrate (Tables 1, 2). Our algorithm is accurate and efficient, which is central to our contribution.
>
> Furthermore, our theoretical results stem from how we characterize the root cause distribution (Definition 1). Unlike prior works (e.g., CF Attribution,[3]) that define root causes using counterfactual contributions, we adopt a probabilistic approach. This approach clarifies several aspects in our RCD procedure. For example:
>
> - Why sampling residual errors from the validation dataset is crucial (line 918).
> - Why fix values must be drawn from conditional distributions (lines 810–822).
>
> *References*
>
> [1] Pawlowski et al., *Deep structural causal models for tractable counterfactual inference*, NeurIPS, 33, pp.857–869.
> [2] Karimi et al., *Algorithmic recourse under imperfect causal knowledge: a probabilistic approach*, NeurIPS, 33, pp.265–277.
> [3] Sharma et al., *The counterfactual-Shapley value: Attributing change in system metrics*, arXiv preprint, arXiv:2208.08399.
>
> > Q1 Abduction as a probability distribution
>
> As mentioned in line 180 of our paper, counterfactuals are point estimates as there is no randomness involved in predicting them, unlike interventions where exogenous variables must be sampled. To draw a probabilistic sampling analogy, we can assume that the learned SCM contains probabilistic functions, such as when each $f_i$ is a Gaussian Process, making $f_i(Pa_{X_i})$ predict a probability distribution over $X_i$ (see [2] above, sec. 4 for an example. Note that the noise $u$ there corresponds to $\epsilon$ in our context). While prior work has shown that such approaches work well in typical CF estimation problem settings, we highlight that this design would also lead to OOD estimation issues in the RCD setup. This is because a root cause test sample could require the posterior over $X_i$ to be predicted from OOD parents. SAGE [4] explores CVAEs to train probabilistic $f_i$s. We request the reviewer to refer to our response to Q7 of reviewer `ERJh` for more details.
>
> *References*
>
> [4] Gan et. Al. Sage: practical and scalable ML-driven performance debugging in microservices. In Proceedings of the 26th ACM International Conference on Architectural Support for Programming Languages and Operating Systems (pp. 135-151).

---

> > ### Author Response · Authors · 2024-11-21
> > **Response 2/n**
> >
> > > Q3 Section 5 is over-explained
> >
> > During the rebuttal, we also felt that this section was verbose. We have significantly trimmed down `Sec. 5` in our revised manuscript.
> >
> > > Q4 (a) Variance of $\epsilon_i$
> >
> > We apologize for the oversight. We had mentioned under “Generating the Oracle SCM” subsection that the $\epsilon_i$ values are sampled from the uniform distribution $U[0, 1]$, resulting in a standard deviation $\sigma(\epsilon_i) = 0.3$. We have now added it to the revised manuscript (`line 447`). In response to your comment, we conducted additional experiments by sampling $\epsilon_i$ from uniform distributions with much broader spreads, which we discuss below.
> >
> >
> > > Q4 (b) Make low variance of exogenous variables" assumption
> >
> > Our method does not explicitly rely on this assumption. However, as our theory shows, high variance in $\epsilon$ can degrade IDI’s performance. High variance in $\epsilon_i$ suggests that the observed parent features do not adequately explain the node’s values, indicating that the dataset lacks strong predictive features. In our experiments, we observed that high variance leads to unstable SCM learning, reflected in high validation errors (see `Appendix E`). This instability affects not only IDI but any method that relies on training an SCM. We elaborate on these aspects in the next question.
> >
> > >Q5  Experiment with high variance of $\epsilon_i$
> >
> > We  start with a toy experiment conducted during rebuttal.
> >
> > **Toy Dataset (`Sec. E` in appendix)**
> >
> > We have updated the anonymized repository to include a notebook (https://anonymous.4open.science/r/petshop-BB8A/limitation.ipynb) that provides a detailed explanation through code. We hope the reviewer finds this helpful.
> >
> > To understand the impact of variance of $\epsilon_i$, we experimented with a simple toy dataset containing four variables, using both linear and non-linear SCMs across four different variance regimes of $\epsilon_i$. Our key observations are as follows:
> >
> > - **Low and Medium Variance Regimes:** Interventional error plateaus at the standard deviation of $\epsilon_i$, while CF errors decrease as the training size increases. Notably, CF errors are heavily correlated with the test errors assessed on OOD root cause samples.
> >
> > - **High Variance Regimes:** As the variance of $\epsilon_i$ increases, the stability of SCM training is impacted, leading to high in-distribution validation errors. This is especially pronounced in high-variance, low-sample scenarios where the learned SCMs are imperfect, negatively affecting both CF and intervention estimates. While CFs outperform interventions in convex linear SCMs (where a global optimum exists), they struggle with more complex non-linear SCMs due to finite sample training behaviors.
> >
> > - **Very High Variance Regimes:** In low-data settings, interventions outperform CFs by a significant margin. However, in extreme variance with high-data scenarios, this trend reverses, favoring CFs.
> >
> > These results highlight the nuanced interplay between variance, data availability, and SCM complexity.
> >
> > **Synthetic Dataset**
> >
> > We experimented with broader exogenous distributions ($U[0, b]$), varying $b$ across $\{0.5, 2, 3, 5\}$ during rebuttal. Please refer `Sec. E` for a detailed answer.  Our empirical observations include:
> >
> > - **Linear SCM:** Both interventional and counterfactual variants of the proposed method (IDI, IDI (CF)) achieved perfect Recall@1 across all variance settings. While interventions exhibited slightly higher absolute error compared to CFs, both methods maintained a consistent rank order over the predicted root causes, leading to identical recall.
> >
> > - **Non-Linear Invertible SCM:** IDI outperformed IDI (CF) in cases with unique root causes but lagged behind by a small margin in test cases involving multiple root causes.
> >
> > - **Non-Linear Non-Invertible SCM:** All CF-based approaches, including IDI (CF), performed very poorly. In contrast, IDI remained robust and outperformed all baselines across different variance settings by a significant margin.
> >
> > **Summary**
> > Across both toy and synthetic settings, interventions demonstrated superior performance in most cases. CF estimates occasionally outperformed in high-sample and extreme variance scenarios. However, when the invertibility of the SCM does not hold, interventions emerged as the clearly superior approach.

---

> ### Author Response · Authors · 2024-11-21
> **Response 3/3**
>
> > Q6 (a) Recall values close to 1
>
> This was true for Linear datasets because linear functions generalize well to anomalous distributions allowing for efficient inference of the fix condition. We did see a significant dip in the recall for the most complex of our SCMs – non-linear and non-invertible.  Regarding Petshop, the temporal cases were easy for baselines (Traversal methods achieve 1) and IDI because they were simulated by injecting too many requests at the root cause node, making the root cause nodes statistically stand out in the test cases.
>
> >Q6 (b) Details of Datasets
>
> We have included this in our revised manuscript at `Sec C`.
>
> > Q6 (c) Report the Std Deviation of Recall
>
> We chose not to report the std deviation as our tables are already quite dense. Many prior RCD works, such as BARO (https://arxiv.org/pdf/2405.09330), TOCA, or CF Attribution also do not include error bars. Further, the trend between high-performing and low-performing methods was clearly visible in our tables. If the reviewer prefers, we are open to include them.
>
> >Q6 (d): Bias Preferring IDI in Dataset
>
> First, we note that the petshop dataset was not created by us, and IDI performs best when compared across the state-of-the-art baselines. For the synthetic datasets that we created, we believe the only bias in our dataset stems from enforcing that the synthetic graph satisfies Assumption 1. We had also conducted an ablation study in `Sec. 6.3` to assess the implications of this assumption. Otherwise, our synthetic functions $f_i$s use random MLPs which are more general. We provide all details on our datasets in `Sec. C` in our revised paper. Further, we have explicitly added limitations in `Sec. 7`.
>
> > Q7 (a) Case studies for low variance
>
> Please refer `Q5` above.
>
> > Q7 (b) typically small in practice
>
> We believe that this statement is in general true for a dataset with rich predictive features. However, we agree that this is unverifiable given training data. For example, in Petshop with latent exogenous variables and unknown Oracle SCM, we cannot comment on the variance($\epsilon$). Therefore, this phrase becomes subjective, and so we removed it.
>
> > Q8 Reduce verbosity
>
> We made several passes over the paper to trim down the redundant points and added limitations.
>
> > Q9
>
> - **Theorem numbers:** We apologize for this. The theorem labels were inadvertently carried over to the statements in the appendix. We corrected this now. Thanks for pointing it out.
> - **Notations** We fixed the symbol $D^{trn}$ and expanded IDI in abstract.
>
> > Q10 Prior methods Sec 3.3
>
> The Hierarchical RCD, CF Attribution, and other traversal approaches support this claim. However, we found this line to be redundant and removed it.
>
> > Q11 remark on page 6
>
> We have revised this remark and removed the reference to CIRCA. Thanks for this suggestion.
>
> > Q12 Z-Score
>
> We have included in our revised paper.
>
> # Questions
>
> > Q1 Auto-regressively
>
> Apologies for the confusion. We should not have used that phrase. In our synthetic data, we generated i.i.d. samples by forward sampling in topological order. We have corrected this throughout the paper.
>
> > Q2 Is Additive noise only way for inverting $\epsilon$
>
> The use of additive noise is indeed the most popular choice, and so we adopted it in our work. However, more sophisticated models exist. As an example, the PostNonLinear (PNL) model [5] provides an alternative. In this model, each $X_i = g(f(\text{Pa}_i) + \epsilon_i)$, where $g$ is a complex but invertible function, and $f$ is arbitrary.
>
> [5] Zhang et. al. On the identifiability of the post-nonlinear causal model. arXiv:1205.2599. 2012 May 9.
>
> > Q3 fixes to any subset of
>
> We have enhanced the clarity in our revised manuscript.
>
> > Q4 Disrupted mechanism/abnormal $\epsilon_j$
>
> Yes, the reviewer’s understanding is correct. For a root cause $X_j$, since the parents are not anomalous, the OOD nature of $X_j$ can only stem from an abnormal $\epsilon_j$.
>
> > Q5 Important Question
>
> We address this question for additive noise models:
> - **Interventions:** No, interventions will not converge to the true counterfactual. In the worst case, the estimation error would plateau at the standard deviation of $\epsilon$ (refer `Appendix E`).
> - **Counterfactuals:** Yes, under ideal conditions. When the learned $\hat{f_i}$ is perfect—e.g., in the limit of infinite training data and when $\hat{f_i}$ consistently estimates the true $f_i$—the counterfactual estimates converge to the true values.
> Thank you for this insightful question; we have incorporated this as a limitation in the revised manuscript.
>
> > Q6 Cloud KPIs are usually positive
>
> Cloud KPIs are typically positive as they measure performance and reliability, including response time, query latency, throughput, CPU utilization, etc. We have added the reference to our manuscript. Notably, IDI does not need positivity of KPI. We explored using Gaussian noise in the toy experiment conducted during rebuttal, as discussed in Q6.

---

> > ### Author Response · Authors · 2024-11-24
> > **Requesting feedback on our rebuttal**
> >
> > Dear Reviewer 7kui,
> >
> > Thank you for your constructive feedback on our paper. With the ICLR public discussion phase ending shortly, we wanted to confirm whether our responses have sufficiently addressed your concerns. If there are any remaining issues, we would be happy to provide additional clarifications.
> >
> > Thank you very much for your time!

---

> > > ### Comment · Reviewer_7kui · 2024-11-24
> > > **Feedback on Rebuttal**
> > >
> > > Thank you for your detailed responses. I re-read the paper after your revisions.
> > > I have the following concerns:
> > >
> > > ## 1) Empirical Evaluation of the Paper's Hypothesis
> > > - The key hypothesis of the paper is that the "*proposed IDI approach outperforms counterfactual estimation methods whenever the variance of the latent exogenous variable $\epsilon_i$ is low*". The authors tried to validate this hypothesis theoretically using Theorems 5 and 6. However, the experimental results do not empirically validate this.
> > >
> > > - Figures 6, 7, and 8 in Appendix E suggest that the proposed IDI approach outperforms counterfactuals in the "**non-linear models**." The counterfactual performs better than the IDI approach in the "linear model". Some exceptional cases lie in low training data size, which can directly result from the law of large numbers (since counterfactual uses "sample mean").
> > >
> > > ## 2) Theoretical Comparison
> > > - I believe the **number of nodes** also has a role to play, that the authors ignored. This is because the first terms of Theorem 5 and Theorem 6 are different (Theorem 6 doesn't have a $2^{n + i -1}$ factor).
> > >
> > > - **Suggestion:** Since the empirical results do not directly validate the theorems in the current version of the paper, the authors must run an experiment on a toy dataset where they explicitly compute each term of Theorem 5 and Theorem 6 to compare what differs between their proposed IDI approach and a counterfactual estimate. The current hypothesis of low variance of $\epsilon_i$ is not the only factor and more investigation is needed to dissect the two theorems in experiments.
> > >
> > > ## 3) Petshop Dataset
> > > - I am not sure why the authors chose the "Petshop" dataset for their experiments. First, the Oracle SCM is missing for Petshop dataset. Secondly, the cases of **(Low, High, Temporal) latency** are not well explained i.e., how latency affects the proposed IDI approach is not apparent from the theoretical description in the paper.
> > >
> > > - If the authors want to demonstrate the application of their proposed IDI approach to a "real-world dataset" and have access to only "Petshop" data for that purpose then the authors should do a few things: (1) Include Petshop experiment at the end of the paper, (2) Provide detailed analysis of how latency affects their approach (in the appendix), and (3) mention that "Oracle SCM is not available for Petshop data thus limiting insights on the applicability of the methods".
> > >
> > > - Out of curiosity, I want to know: Is Petshop the benchmark dataset in the literature? If yes, is there any literature which has tried to estimate an oracle SCM for the data ?
> > >
> > > ## 4) Minor Issue
> > > - The issue with the formatting of the paper i.e., the numbering of theorems was raised in the official comment. However, the problem persists. The main text numbers the theoretical results as "Theorem 5", "Theorem 6", "Corollary 7". Appendix B numbers it as "Theorem 10", "Theorem 11", "Corollary 12".
> > >
> > >
> > > I will keep my score as is.
> > > I am open to responses during the discussion period.

---

> ### Author Response · Authors · 2024-11-25
> **Response to the Rebuttal feedback**
>
> Dear Reviewer,
>
> Thanks very much for your time and effort in reviewing our work. Please see below for the questions raised during the rebuttal.
>
>
> # Empirical validation of Paper’s Hypothesis
>
> We reiterate, as stated in line 77 of our paper, that if the oracle values of $\epsilon$ are available, counterfactuals (CFs) are the optimal choice for RCD. The linear experiments in our paper in Figures 6, 7, 8 exemplify this *ideal* scenario, since the estimated $\hat{f}_i$ converges to the true $f_i$ in the Oracle SCM. We have clearly highlighted this in our conclusion in Sec. 7 of our revised paper, emphasizing the limitation that CFs outperform interventions when the estimated SCM closely approximates the true SCM, i.e., $\hat{f}_i \approx f_i$.
>
> Moreover, in our proof, if additional assumptions are made about the gap between $f_i$ and $\hat{f}_i$ is small, the CF error analysis in Equations 10 and 11 could achieve much tighter bounds. In the absence of such knowledge, the goal of Theorem is to provide bounds for more general scenarios.
>
> Additionally, we want to emphasize that the linear experiments presented in our work do not conclusively suggest that CFs will always outperform interventions in all linear settings. It is possible to construct examples where the Oracle SCM uses linear functions, yet even with extensive training data, the estimates $\hat{f}_i$ may exhibit low error in the training regime but high error when applied to the abnormal root cause regime. One such case is multicollinearity, which we explain in more detail below:
>
> To explicitly show the dependence of the CF error on the gap between the prediction errors during training regime and the anomalous root cause regime, we extended the four-variable toy linear dataset presented during the rebuttal as follows:
>
> - The input features were expanded to six dimensions. The first three features, $X[:, 0:3]$, were sampled i.i.d. from a standard Gaussian as earlier. The remaining three features were made correlated with the first three as follows:
>   $$
> X[:, 4] = X[:, 1] + \mathcal{N}(0, \rho), \quad X[:, 5] = X[:, 2] + \mathcal{N}(0, \rho), \quad X[:, 6] = X[:, 3] + \mathcal{N}(0, \rho)
>   $$
>   Here, $\mathcal{N}$ is Gaussian distribution, and $\rho$ -- its variance takes values in \{0.01, 0.1, 0.5, 1\}. When $\rho = 0.01$, the fourth column is heavily correlated with the first, while for $\rho = 1$, the correlation coefficient is less pronounced.
>
> - We considered low training size with 50 samples and high training size with 1000 samples to study the impact of such correlated features on the intervention and counterfactual errors. We worked with the very high variance setting var($\epsilon_4$)=5, where interventions demonstrated poor performance earlier.
>
>
>
> We have updated the anonymized repository with the notebook for this specific experiment. The notebook can be accessed at the following link: https://anonymous.4open.science/r/petshop-BB8A/limitation_linear.ipynb.
>
> The results (Section E.1, the new figures 6e, 6f) reveal the following:
> 1. For high training size and high variance settings:
>    - First, observe that the validation error remains close to zero across all $\rho$ settings, but the root cause test error blows up in the abnormal regime for smaller $\rho$s. This elucidates that even linear functions can have high out of domain error.
>    - In high training size case, we observed that when $\rho$ is small, interventions outperform CFs because multicollinearity induces high error in the abnormal root cause regime for CFs.
>    - As $\rho$ increases (e.g., $\rho > 0.5$), the effects of multicollinearity diminish, and CFs regain their dominance over interventions.
>
> 2. For low training size across all $\rho$ values:
>    - Interventions consistently outperformed CFs. CF error escalates to as high as $10^3$, while interventions remain robust with their error bounded by the standard deviation of $\epsilon$.
>
> Thus, our observations in the toy dataset align with our theoretical analysis. The findings are not contradictory but highlight the nuanced conditions under which CFs or interventions dominate.
>
> We appreciate the reviewer’s acknowledgement that in complex non-linear cases, IDI generally outperforms CFs.
>
> - **General Insights:**
>   The central thesis of our work is that while there are specific scenarios where counterfactuals (CFs) outperform interventions—such as when the abducted values are accurately estimated and closely align with the oracle values (e.g., when $\hat{f}_i \approx f_i$)—these cases are exceptions rather than the norm. When consolidating the performance across the diverse SCMs examined in our study, interventions surely emerge as the superior approach for RCD compared to CFs.

---

> ### Author Response · Authors · 2024-11-25
> **Response to Rebuttal feedback**
>
> # Number of Nodes
>
> The bounds in our theoretical results are not affected by the total number of nodes in the SCM. Instead, they are determined by the number of nodes in the paths between the root cause node and the target. This is because our bounds include $n-i$ in the exponent, where $n$ represents the total number of nodes in the chain graph, and $i$ is the position of the root cause node. Consequently, $n-i$ corresponds to the path length from the root cause node $i$ to the target node $n$. Thank you for this question—it highlights an important experiment to demonstrate how the depth of the anomalous root cause node impacts the estimates of CFs and interventions.
>
> We report the errors in the estimated CF values and the estimated interventional values when compared to the true CF values at nodes located at different depths from the root cause node. These results are computed for an example test case using a non-linear invertible synthetic dataset. We compute the percentage reduction in error is defined as:
>
> $$
> \text{Reduction} = \left(\frac{\text{CF Error} - \text{Int Error}}{\text{CF Error}}\right) \times 100
> $$
>
> The table below presents the errors for CFs and interventions (int) against their depth from the root cause node, alongside the percentage reductions:
>
> | Depth from Root Cause Node | CF Error | Int Error | % Reduction in Error |
> |----------------------------|----------|-----------|-----------------------|
> | 1                          | 0.07     | 0.05      | 38.13%               |
> | 2                          | 0.19     | 0.12      | 37.47%               |
> | 3                          | 0.54     | 0.39      | 27.75%               |
> | 4                          | 0.78     | 0.65      | 16.52%               |
> | 5                          | 1.51     | 0.88      | 41.76%               |
> | 6                          | 2.32     | 1.24      | 46.77%               |
> | 7                          | 5.45     | 1.55      | 71.55%               |
>
> We observed similar trends across various test cases. As the depth from the root cause node increases, the CF estimates show greater errors. However, the interventional estimates demonstrate a significant advantage, maintaining lower errors.
>
>
>
> # Petshop
>
> **Is petshop a benchmark dataset from literature?**
>
> Yes, Petshop is a dataset recently introduced in [Hardt et al., 2023]. We did not create it. It contains root cause test cases created by manually injecting faults into a real-world micro service application. This dataset addresses a key research gap in other RCD datasets by making the underlying dependency graph publicly available, which serves as the causal graph.
>
> **Why petshop?**
>
> This dataset focuses on root cause analysis of real-world anomalies occurring on a deployed website called Petshop. The target node in this dataset where anomalies are triggered is PetSite—the front-end interface where users interact to browse for the pets displayed in the website. An anomaly is triggered at Petsite upon a violation of the service level objective (SLO), such as when the website's response time exceeds 200 microseconds. Root cause test cases were manually injected by selecting a node and issuing an abnormally high number of requests through that node. This surge in traffic from the selected root cause node ultimately triggers an SLO violation at the Petsite node.
>
> The dataset includes three types of root cause test cases: **Low**, **High**, and **Temporal** latency. On average, the request rates were 485 requests per second for Low latency, 690 for High latency, and 571 for Temporal latency cases. In the High and Temporal latency cases, the root cause nodes deviated significantly from the request patterns observed during training, allowing baseline methods such as traversal to perform well. However, for the more challenging Low latency cases, where the statistical discrepancy of the root cause nodes between the training regime and the abnormal root cause regime is less pronounced, IDI demonstrated the best performance.
>
> **Oracle Causal Graph vs. Oracle SCM**
>
> While the Oracle causal graph is available in Petshop, the Oracle SCM remains unavailable since the true structural equations $f_i$ underlying the Oracle SCM are unknown. To the best of our knowledge, no prior work has explicitly defined the Oracle SCM. We argue that obtaining the exact Oracle SCM (i.e., the precise $f_i$ functions) is infeasible for any real-world dataset, and rather only observed samples can be recorded. Previous studies leveraging Petshop, such as [TOCA], have adopted a learning-based approach, approximating the Oracle SCM by training the structural equations $\hat{f_i}$ on the collected finite samples. Our work also followed this same methodology.
>
>
> # Minor Issue
> We had different numbers to ensure the continuity of numbers in the Appendix. We have now updated the theorem numbers.

---

> > ### Comment · Reviewer_7kui · 2024-11-26
> > **Response to Rebuttal**
> >
> > Thank you for addressing the clarifications and conducting the additional experiments. I have a few further suggestions for improving the paper:
> >
> > (1) I recommend modifying the sentence, “*Our theoretical analysis demonstrates that IDI’s in-distribution intervention approach outperforms other counterfactual estimation methods whenever variance of the underlying latent exogenous variables is low*,” to instead highlight that the paper performs a comparative evaluation in scenarios where IDI outperforms counterfactual estimation methods. Additionally, consider removing the phrase “variance of the underlying latent exogenous variables is low” from the main text.
> >
> > While I initially suggested including it, the nuanced results on "linear" vs. "non-linear" cases and "multicollinearity" highlight that readers (particularly practitioners) should be able to assess both methods' strengths and weaknesses and make an informed choice between counterfactual estimation and the IDI approach.
> >
> > (2) Please provide a theoretical explanation for the observation that “*high $\rho$ makes counterfactual estimation outperform the proposed IDI approach.*” This insight will be valuable to readers.
> >
> > (3) If possible, repeat the “depth of the root cause node” experiment for the *linear, high training data, high $\rho$* setting. While I acknowledge that counterfactual estimation might outperform IDI in this scenario, it would be helpful to identify any boundary cases where IDI can still outperform.
> >
> > Please include this experiment in the paper (e.g., the appendix), as it is likely to be of significant interest to potential readers.
> >
> > (4) Include the following key assumption explicitly in the paper: “The oracle values of the latent exogenous variables are unknown a priori.”
> >
> > (5) Please include a detailed description of the Petshop dataset in the appendix, covering the anomaly, root cause, and latency aspects you mentioned earlier.
> >
> > Overall, I believe the paper provides a thorough empirical evaluation of the IDI approach, effectively comparing it with state-of-the-art counterfactual estimation methods. With these updates, I am happy to raise my score.

---

> > > ### Author Response · Authors · 2024-11-28
> > > **Response to Rebuttal feedback**
> > >
> > > Thanks for the rebuttal feedback. We highly value the reviewer's engagement with our work. Please find our responses below:
> > >
> > > > 1a. Modify the sentence to include a comparative evaluation
> > >
> > > Thank you for the suggestion. We have revised the abstract to state:  "We present a theoretical analysis comparing and bounding the errors in assessing the fix condition using interventional and counterfactual estimates. We then conduct experiments by systematically varying the SCM’s complexity to demonstrate the cases where IDI’s interventional approach outperforms the counterfactual approach and vice versa."
> > >
> > > Please let us know if this modification is fine.
> > >
> > > > 1b. Remove “variance of the underlying latent exogenous variables is low”
> > >
> > > We have removed the phrase in our revised uploaded paper. However, we have retained it in line 317 for brevity while explaining the interventional error bound.
> > >
> > > > 2. Theory for high $\rho$
> > >
> > > We explain this with an example. Suppose, we consider an oracle function $y = 3x$. If we estimate this function using $2d$ covariates $[X_1, X_2]$ with perfect correlation $ X_1 = X_2 = x$, the regression coefficients $\hat{w}_1$ and $\hat{w}_2$ in $\hat{y} =  \hat{w}_1 x_1 +  \hat{w}_2x_2$ can become highly unstable. Because, the problem has several valid solutions like $[3, 0], [0, 3]$, or $[100, -97]$. This unstability exacerbates the estimation errors especially when they are evaluated on OOD inputs. We refer the reviewer to [1, Chapter 9] for more details. We will try to include a formal proof in the revised paper.
> > >
> > >  [1] Chatterjee, S. and Hadi, A.S., 2015. Regression analysis by example. John Wiley & Sons.
> > >
> > > > 3. Depth of the root cause experiment
> > >
> > > This is indeed a valuable experiment. Unfortunately, we could not complete this experiment before the PDF revision deadline. We will present the results before the rebuttal deadline and include the results in the camera-ready version if accepted.
> > >
> > > > 4. Assumption that oracle $\epsilon$ is unknown
> > >
> > > It is widely accepted that $\epsilon$ values are unobservable. Hence, we prefer not to include this as an explicit assumption. However, we have mentioned it in line 79 for clarity. Please let us know your thoughts on this.
> > >
> > > > 5. Detailed description of the Petshop dataset
> > >
> > > We have included the dataset details in Appendix C.1.
> > >
> > > > Paper Revision
> > >
> > > We are grateful for the reviewer’s many insightful suggestions. In particular, we believe the toy experiment insights should be moved to the main paper, as these simple settings effectively illustrate our core contributions. We will ensure this is included in subsequent revisions.
> > >
> > > If there are any further clarifications or suggestions, we would be happy to address them.

---

> > > > ### Author Response · Authors · 2024-12-02
> > > > **Depth of the Root Cause Experiment**
> > > >
> > > > Dear Reviewer,
> > > >
> > > > Thank you once again for engaging closely with us during the rebuttal process.
> > > >
> > > >
> > > > > **Depth of the Root Cause Node Experiment**
> > > >
> > > > We experimented with a linear model, high-variance ($5$), and high training data ($1000$ samples). We provide results using two settings of $\rho$: Low level of multicollinearity ($\rho=1.0$) and high levels of multicollinearity ($\rho=0.01$).
> > > >
> > > > **Dataset Details**: We considered a causal graph with the root containing six nodes. The first three nodes were sampled from a standard Gaussian $\mathcal{N}(0, 1)$, while the fourth node was correlated with the first node as $X[1] + \mathcal{N}(0, \rho)$, and so on.
> > > >
> > > > The causal graph has a depth of $10$ levels, with six nodes per level for the first nine levels. At each depth $d \in \{1, 2, \ldots, 9\}$, the nodes are defined as a linear transformation from the nodes at their previous depth, as follows:
> > > > $$
> > > > X_d = \frac{X_{d-1} \mathbf{W}_{d}}{d} + \epsilon
> > > > $$
> > > >
> > > > where $\mathbf{W}_d \in \mathbb{R}^{6 \times 6}$ is a linear transformation matrix with coefficients sampled from $\mathcal{N}(0, 1)$, and $\epsilon \in \mathbb{R}^6$ are the exogenous variables sampled from $\mathcal{N}(0, 5)$. The value $5$ reflects the `high-variance` setting. In the last level, we have a single node $Y$ defined as a linear function of the nodes in last before layer.
> > > >
> > > > The updated notebook for this experiment is available here: [limitations_linear_depth.ipynb](https://anonymous.4open.science/r/petshop-BB8A/limitations_linear_depth.ipynb). The results for each $\rho$ across all depth levels are reported below.
> > > >
> > > > | Depth  | $\rho = 0.01$              |                     |             | $\rho = 1.00$            |                     |             |
> > > > |--------|------------------------------------|---------------------|-------------|-----------------------------------|---------------------|-------------|
> > > > |        | Int Errors                         | CF Errors           | % Reduction | Int Errors                        | CF Errors           | % Reduction |
> > > > | 1      | 5.28                               | 82.22              | 93.58       | 5.30                              | 5.53                | 4.16        |
> > > > | 2      | 5.97                               | 84.59              | 92.94       | 5.98                              | 6.20                | 3.55        |
> > > > | 3      | 6.01                               | 83.83              | 92.83       | 6.00                              | 6.12                | 1.96        |
> > > > | 4      | 5.63                               | 82.28              | 93.16       | 5.64                              | 5.76                | 2.08        |
> > > > | 5      | 5.55                               | 83.58              | 93.36       | 5.56                              | 5.81                | 4.30        |
> > > > | 6      | 5.83                               | 83.09              | 92.98       | 5.83                              | 6.06                | 3.80        |
> > > > | 7      | 6.58                               | 85.21              | 92.28       | 6.59                              | 6.90                | 4.49        |
> > > > | 8      | 5.84                               | 82.80              | 92.95       | 5.83                              | 6.05                | 3.64        |
> > > > | 9      | 6.56                               | 84.29              | 92.22       | 6.56                              | 6.80                | 3.53        |
> > > > | 10     | 7.17                               | 86.70              | 91.73       | 7.18                              | 7.56                | 5.03        |
> > > >
> > > > We make the following observations:
> > > >
> > > > 1. The intervention errors plateau at the standard deviation of $\epsilon$ at each depth level, across both settings of $\rho$. However, counterfactual (CF) errors are comparable to intervention errors only when $\rho$ is large. For smaller $\rho$, CF errors degrade significantly.
> > > > 2. Unlike the non-linear case, we do not observe an increasing trend in the percentage reduction with depth. This is due to the high training size and the inherent property of linear models, where their functional behavior remains consistent across usual training and abnormal root cause regimes.
> > > > 3. When we repeat the experiments with a smaller training size (50 samples), an increasing trend in percentage reduction emerged.
> > > >
> > > > These results further emphasize that in scenarios where we lack certainty about how closely the learned $\hat{f_i}$s approximate the true $f_i$s, interventions are a preferable choice.
> > > >
> > > > We would greatly appreciate the reviewer's feedback on these findings and are eager to continue the discussion. Thank you once again!

---

> > > > > ### Comment · Reviewer_7kui · 2024-12-02
> > > > >
> > > > > Thank you for the additional experiments.
> > > > >
> > > > > (1) All your experiments prove that the notion of superior performance between CF and the proposed IDI approach is nuanced. There's an interplay between correlation, depth of the node, the variance of exogenous variables, training data size, etc.
> > > > > At this point, I am not sure if there is a clear conclusion to these studies apart from being a comparative analysis between the two approaches. The reason behind choosing the IDI approach needs further motivation using real-world examples (engineering, biomedical, and econometrics applications).
> > > > >
> > > > > (2) I previously stated to mention the assumption that oracle $\epsilon$ values are unknown. My apologies. I overlooked the fact that it is termed as a ``latent exogenous variable''. I take back my instruction on that part. Thank you for pointing that out.
> > > > >
> > > > > Overall, I am very impressed by how the paper has evolved throughout the discussion period. Thanks a lot for taking the reviews in a constructive mindset and working on further experiments. I will retain my positive score.

---

### Official Review · Reviewer_hq7T · 2024-11-04

**Soundness:** 3
**Presentation:** 3
**Contribution:** 2
**Rating:** 6
**Confidence:** 2

**Summary:**

This paper focuses on addressing the issue of performance degradation caused by OOD problems in previous methods when localizing root cause curves through causal interventions. It proposes the IDI method and demonstrates its advantages over existing SCM-based root cause localization methods through multi-level experiments.

**Strengths:**

1. The presence of hidden confounding factors makes the OOD problem both prevalent and significant.
2. The writing of this paper is clear and easy to read.
3. The paper provides proof that the IDI method's in-distribution sampling has a bounded error, scaling with the distance between anomalies and normality.

**Weaknesses:**

See the following questions.

**Questions:**

1. The paper uses intervention methods to identify the root cause curve from multiple anomalous KPI curves. MicroCause uses Granger causality. Can there be a comparative result between the two? [1].
2. On line 345, it is mentioned that to eliminate the impact of OOD, modeling needs to be performed on all subsets of candidates. What is the computational overhead when a fault is associated with many anomalies?

[1] Meng, Yuan, et al. "Localizing failure root causes in a microservice through causality inference." 2020 IEEE/ACM 28th International Symposium on Quality of Service (IWQoS). IEEE, 2020.

---

> ### Author Response · Authors · 2024-11-20
> **Response 1/n**
>
> Thank you very much for appreciating our work and providing positive feedback. We are happy to engage with you for any follow-up questions.
>
> > Q1 Comparison with MicroCause
>
> Thank you for pointing us to this prior work. The referenced paper (MicroCause) addresses the challenge of learning causal graphs for microservices datasets. MicroCause highlights that conventional graph discovery methods such as the PC algorithm, struggle to recover the true causal graph in RCD settings. To address this, the authors introduced a more sophisticated PCTS algorithm for causal graph learning. However, once the graph is learned, MicroCause employs a random walk *traversal* approach for root cause identification. In comparison, since the main focus of our work is to analyze and develop an algorithm for root cause prediction, we assumed access to the true dependency graph of the components. Therefore, the causal graph learning algorithm of MicroCause is complementary to our approach. Their root cause identification algorithm is a variant of the traversal-based approaches discussed in our paper (Sec. 6). This paper can thus be categorized as part of the traversal-based family of approaches. We have cited and discussed this paper at lines 114, 376 in our revised manuscript.
> We report the comparison with Traversal methods run on the true causal graph for reference on the petshop dataset.
>
> | Recall@ | Method            | Low (k=1) | Low (k=3) | High (k=1) | High (k=3) | Temporal (k=1) | Temporal (k=3) |
> |---------|-------------------|-----------|-----------|------------|------------|----------------|----------------|
> |         | Traversal         | 0.80      | 0.80      | 0.90       | 0.90       | 1.00           | 1.00           |
> |         | Ours (IDI)       | 0.90      | 0.90      | 0.90       | 0.90       | 1.00           | 1.00           |
>
>
>
> > What is the computational overhead when a fault is associated with many anomalies?
>
> We assessed the running time of all the baseline used in our work. Please see below for the results. We request the reviewer to also refer to Sec. D in Appendix for a detailed answer.
>
> **Running Time for Unique Root Causes**
>
> 1. The Correlation and Causal Anomaly methods demonstrate the best performance in terms of running time.
> 2. Causal Fix approaches are bottlenecked by the need to learn the Structural Causal Model (SCM), with linear methods taking less time compared to non-linear ones due to the latter's reliance on gradient-based training.
> 3. Ours (CF) and Ours do not require Shapley value computations for predicting unique root causes, significantly reducing running time.
> 4. The CF Attribution method performs Shapley analysis across all nodes, making it the slowest method.
>
>
> *Table: Running time (in seconds) for datasets with **unique** root causes. "--" indicates that the baseline was not executed on the corresponding dataset due to incompatible assumptions.*
>
> | Category         | Method                | Petshop | Syn Linear | Syn Non-Linear |
> |------------------|-----------------------|---------|------------|----------------|
> | Correlation      | Random Walk          | 2.36    | 1.74       | 4.81           |
> |                  | Ranked Correlation   | 0.60    | 0.21       | 2.99           |
> |                  | ε-Diagnosis          | 2.11    | --         | --             |
> | Causal Anomaly   | Circa                | 0.52    | 0.36       | 2.73           |
> |                  | Traversal            | 0.27    | 0.24       | 1.05           |
> |                  | Smooth Traversal     | 0.30    | 0.26       | 0.99           |
> | Causal Fix       | HRCD                 | 11.69   | --         | --             |
> |                  | TOCA                 | 1.96    | 0.95       | 9.16           |
> |                  | CF Attribution       | 9.71    | 22.99      | 178.47         |
> | Ours             | IDI(CF)            | 0.42    | 0.38       | 8.31           |
> |                  | IDI                 | 0.37    | 1.29       | 9.62           |

---

> > ### Author Response · Authors · 2024-11-20
> > **Response 2/2**
> >
> > **Running Time for Multiple Root Causes**
> >
> > 1. The random walk-based approach has increased running time due to multiple anomalous paths.
> > 2. All other baseline methods exhibit running times similar to unique root cause cases.
> > 3. IDI and IDI (CF) require Shapley analysis in this setting but compute Shapley values only for a subset of nodes, leading to a significantly smaller increase in running time compared to CF Attribution.
> >
> > *Table: Running time (in seconds) for datasets with **multiple** root causes.*
> > Observations:
> > | Category         | Method                | Syn Linear | Syn Non-Lin Inv | Syn Non-Lin Non-Inv |
> > |------------------|-----------------------|------------|------------------|---------------------|
> > | Correlation      | Random Walk          | 8.63       | 6.73             | 9.10                |
> > |                  | Ranked Correlation   | 0.26       | 1.54             | 1.90                |
> > | Causal Anomaly   | Circa                | 0.38       | 2.34             | 2.23                |
> > |                  | Traversal            | 0.22       | 2.26             | 1.99                |
> > |                  | Smooth Traversal     | 0.26       | 2.37             | 1.95                |
> > | Causal Fix       | TOCA                 | 1.30       | 11.66            | 13.58               |
> > |                  | CF Attribution       | 23.48      | 120.61           | 190.08              |
> > | Ours             | IDI(CF)            | 7.08       | 40.12            | 73.41               |
> > |                  | IDI                | 8.20       | 42.60            | 76.24               |
> >
> >
> >
> >
> >
> > ---
> >
> > **Number of Nodes Considered for Shapley Analysis**
> >
> > To assess the complexity of Shapley values used in IDI, we compare it against CF Attribution baseline which also uses Shapley values.
> >
> > Observations:
> > 1. Shapley value computations are NP-Hard, often requiring Monte Carlo approximations. We used 500 permutations for all methods to ensure tractability.
> > 2. Ours achieves a 6× reduction in the number of nodes compared to CF Attribution, significantly lowering the computational cost. The reduction would be even more pronounced with exact Shapley computations.
> >
> >
> > *Table: Number of nodes considered for Shapley analysis (mean ± standard deviation across all test cases).*
> >
> > | Dataset                  | CF Attribution       | Ours                 |
> > |--------------------------|----------------------|----------------------|
> > | Linear                   | 36.4 ± 3.6          | 6.0 ± 1.6           |
> > | Non-Linear Invertible    | 35.8 ± 4.2          | 6.0 ± 1.9           |
> > | Non-Linear Non-Invertible| 37.1 ± 3.8          | 6.2 ± 1.9           |

---

> > > ### Author Response · Authors · 2024-11-24
> > > **Requesting feedback on our rebuttal**
> > >
> > > Dear Reviewer hq7T,
> > >
> > > Thank you for your constructive feedback on our paper. With the ICLR public discussion phase ending shortly, we wanted to confirm whether our responses have sufficiently addressed your concerns. If there are any remaining issues, we would be happy to provide additional clarifications.
> > >
> > > Thank you very much for your time!

---

> > > > ### Comment · Reviewer_hq7T · 2024-11-26
> > > >
> > > > Thank you for your detailed reply. In practical scenarios, if we have the ground truth dependency graph, root cause analysis or prediction is not as challenging. Based on my current judgment, this paper has not yet reached the level to score an 8, so I will maintain my score for now.

---

> > > > > ### Author Response · Authors · 2024-11-28
> > > > > **Response to the Rebuttal feedback**
> > > > >
> > > > > We thank the reviewer for reviewing our rebuttal and providing valuable feedback.
> > > > >
> > > > > We believe RCD remains a challenging problem, even under the assumption of a dependency graph. This challenge arises because RCD often involves test cases where the root cause nodes and their descendants possess *OOD* values. Consequently, the predominant counterfactual-based approaches for RCD perform OOD evaluations using their trained models. Our paper’s primary contribution lies in analyzing and bounding the errors introduced by such OOD inputs. These insights led us to rigorously develop and propose our intervention-based approach, IDI, as a more robust alternative.
> > > > >
> > > > > We are eager to engage further with the reviewer and conduct additional experiments to address specific concerns.

---

### Author Response · Authors · 2024-11-21
**Common Response**

**Dear Reviewers,**

Thank you for your comprehensive and constructive feedback on our work. Below, we summarize the key revisions made during the rebuttal:

- **Datasets:**  We clarify that the Petshop dataset is a recently released, state-of-the-art RCD benchmark, and not created by us. Details about Petshop and other synthetic datasets have been added to `Appendix C`.
- **Writing:**    We streamlined the paper by removing redundant points and updated the caption of `Fig. 1` to clarify the OOD behavior specific to the RCD problem. Additionally, we introduced `Fig. 2` to illustrate the training and inference procedures of IDI.

- **Limitations of IDI:**  To analyze the pros and cons of IDI compared to other counterfactual RCD approaches, we conducted new experiments (see `Appendix E`) to evaluate IDI’s sensitivity to training size and variance ($\epsilon_i$). These limitations have been explicitly discussed in `Sec. 7`.

- **Timing Analysis:**   We performed a timing analysis of IDI and other baselines, now included in `Appendix D`.

Thank you once again for your insightful comments.

---

### Meta-Review · Area_Chair_ihiN · 2024-12-20

**Metareview:**

This paper addresses the challenge of diagnosing the root causes of anomalies in complex interconnected systems, such as cloud services and industrial operations. Traditional approaches rely on counterfactual estimates derived from Structural Causal Models (SCMs) trained on historical data to identify root causes. However, these counterfactual estimates are frequently unreliable since anomalies are rare and often out-of-distribution (OOD) events.

To overcome this limitation, the paper introduces In-Distribution Interventions (IDI), a novel algorithm designed to identify root causes based on two criteria: (1) Anomalous Condition – the root cause nodes must exhibit anomalous values, and (2) Fix Condition – had these nodes assumed normal values, the target node would not have been anomalous. Unlike traditional methods, IDI relies on interventional estimates obtained from SCMs probed only within the training distribution, avoiding OOD issues.

Theoretical analysis demonstrates that IDI outperforms existing counterfactual-based methods, particularly when the variance of the underlying exogenous variables is low. Experimental evaluations on synthetic and real-world benchmark datasets, including the Petshop RCD dataset, show that IDI consistently identifies true root causes with greater accuracy and robustness compared to nine state-of-the-art baseline methods.

By leveraging in-distribution interventions, IDI improves the diagnosis of anomalies' root causes. It avoids the pitfalls of unreliable counterfactuals while maintaining high performance across various scenarios. This work provides a practical and effective solution to root cause diagnosis, particularly in complex systems where anomalies are rare and traditional methods fail.

Overall, most reviewers comment positively about the content and contribution of the paper. I concur with the reviewers, and recommend this paper for publication.

**Additional Comments On Reviewer Discussion:**

There have been fruitful discussions among the authors and the reviewers. This has led to the authors successfully addressing most of the criticisms/questions raised by the reviewers.

---

### Decision · Program_Chairs · 2025-01-22

Accept (Poster)